# Effect of Imaging Range on Performance of Terahertz Coded-Aperture Imaging

**DOI:** 10.3390/s25185667

**Published:** 2025-09-11

**Authors:** Yan Teng, Haodong Yang, Xinhong Cui, Xiaoze Li, Yanchao Shi

**Affiliations:** Key Laboratory of Advanced Science and Technology on High Power Microwave, Northwest Institute of Nuclear Technology, Xi’an 710024, China; yanghaodong@nint.ac.cn (H.Y.); cuixinhong@nint.ac.cn (X.C.); lixiaoze@nint.ac.cn (X.L.); shiyanchao@nint.ac.cn (Y.S.)

**Keywords:** terahertz coded-aperture imaging, reference-signal matrix (RSM), column and row correlation, spatiotemporal independence, imaging range

## Abstract

This paper reveals a counterintuitive, non-monotonic dependence of terahertz coded-aperture imaging (TCAI) performance on the imaging range. This phenomenon stems from phase-induced spatiotemporal correlations in the reference-signal matrix (RSM), governed by the wavefront phase interactions between the coded-aperture elements and scatterers on the imaging plane. Image quality deteriorates noticeably when a specific dimensionless criterion, which is defined mathematically and physically in this work, precisely reaches integer values. Under such conditions, the relative phase difference concentrates or clusters into discrete values determined by the imaging range, leading to strong column and row correlations in RSM that compromise the spatiotemporal independence essential for high-quality reconstruction. For imaging ranges exceeding the critical threshold determined by the number of grid points along one dimension of the imaging plane, two degradation mechanisms emerge: increased correlation between RSM columns mapping to directly adjacent scatterers and phase coverage reduction in wavefront encoding. Both effects intensify as the imaging range increases, resulting in a monotonic deterioration of imaging performance. Crucially, reconstruction fails primarily when strong correlations involve dominant scatterers, whereas correlations among non-dominant (dummy) scatterers have a negligible impact. The Two-step Iterative Shrinkage/Thresholding (TwIST) algorithm demonstrates superior robustness under these challenging conditions compared to some other conventional methods. These insights provide practical guidance for optimizing TCAI system design and operational range selection to avoid performance degradation zones.

## 1. Introduction

Recent advances in terahertz (THz) radar imaging take advantage of THz waves’ superior penetration over optical waves and higher resolution than microwaves [1,2]. Notable systems include the United States DARPA-developed 235 GHz Video Synthetic Aperture Radar (ViSAR) for all-weather real-time imaging [3,4], and China’s National University of Defense Technology (NUDT) 216 GHz radar, which achieved 0.18 m resolution in airborne tests for dynamic scene imaging [5]. However, these systems are still limited by conventional synthetic aperture radar (SAR) and inverse SAR (ISAR) methodologies originally designed for centimeter and millimeter wavebands, which require coordinated sensor–target motion and extended integration times to synthesize apertures. This fundamentally limits their efficiency in forward-looking or staring scenarios which demand rapid data acquisition from static platforms.

To address the limitations of SAR-related technologies, terahertz coded-aperture imaging (TCAI) has emerged as a transformative approach integrating optical coded-aperture imaging and radar coincidence imaging [6,7]. It facilitates instantaneous high-resolution forward-looking and staring imaging via spatiotemporally modulated single-terahertz beams, eliminating the need for target-sensor relative motion. Since TACI was first proposed, researchers have made significant advances in both hardware and computation [8,9,10,11,12]. Harvard Robotics Laboratory developed a 1024-element coded aperture subreflector array with 1-bit phase shifters for military applications [13]. Other hardware innovations include single-input–multiple-output architectures [14,15], and incoherent detector arrays, which replace sequential temporal sampling with parallel spatial sampling to enable dynamic target tracking [16]. In computation, challenges from large-scale reference signal matrices and low-SNR 3D scenarios have driven advanced reconstruction algorithms, such as matched filtering-based TCAI [17], sparsity-driven TCAI [18], geometric measure-optimized TCAI [19], backpropagation-enhanced TCAI [20], and deep attention network-assisted TCAI [21]. Recently, deep learning frameworks (e.g., convolutional neural networks) have been integrated to enhance high-resolution 3D imaging under noisy conditions [22,23,24].

Despite these significant advancements, research efforts have predominantly focused on actively enhancing RSM performance through improved hardware design [13,14,15,16] and reconstruction algorithms [17,18,19,20,21,22,23,24], typically under a fixed imaging range *L*. In contrast, the passive, inherent relationship between the imaging range *L* and the spatiotemporal independence of the reference-signal matrix (RSM) has received considerably less attention. Conventional radar theory suggests an inverse relationship between resolution and imaging range. This has led to the natural—yet unexplored—hypothesis that the spatiotemporal independence of the RSM should also scale inversely with *L* [25]. Contrary to this expectation, this work reveals a counterintuitive, non-monotonic dependence of the TCAI performance on the imaging range.

This work specifically addresses this unexplored aspect by quantitatively investigating the effect of the imaging range on the spatiotemporal independence of the RSM and the consequent TCAI performance. To elucidate the physical origin of imaging range dependence by decoupling wavefront phase effects from signal bandwidth effects, this study deliberately excludes the terahertz signal bandwidth from the model, thereby omitting analysis of range resolution. The dependence of TCAI performance on the imaging range is found to be governed by the relative phase relationships between the coded-aperture elements and the imaging-plane scatterers. And these relative phase relationships can be measured by a dimensionless criterion, which provides the key threshold of the imaging range for the application of TCAI. The remainder of this paper is organized as follows. The operational principles and mathematical framework of TCAI are elaborated on in Section 2, establishing the theoretical foundation for subsequent analyses. The effect of the imaging range on the spatiotemporal independence characteristics of RSM is investigated theoretically and quantitatively in Section 3. Then, the theoretical analysis is numerically validated by comprehensive computational experiments in Section 4, followed by a brief discussion and the conclusions, which are presented in Section 5.

## 2. Operation Principle and Mathematical Framework

### 2.1. Theoretical Model and Signal Transmission

The forward-looking TCAI system comprises a master controller, data processor, transmitter, receiver, and two coded apertures A and B, as depicted in Figure 1. This system synchronously encodes both transmitted and echo signals, enhancing resolution beyond single-terminal encoding approaches [26]. Coded aperture A, which is positioned at the transmitting terminal, and aperture B at the receiving terminal are both controlled by the master controller. Both coded apertures A and B randomly modulate the amplitude or phase of the incident terahertz wavefront in different bit-depths. The transmitter emits single-terahertz wave beams in periodic pulses at fixed sampling intervals *T_s_*. Coded aperture A applies random encoding to the transmitted signal, generating spatiotemporally uncorrelated radiation patterns. After interacting with scatterers on the imaging plane, echo signals are randomly encoded by coded aperture B and collected by the receiver. The data processor reconstructs the target image by solving the inverse problem formulated as a matrix equation incorporating the echo-signal vector, the large-scale RSM, and noise components.

As shown in Figure 1, at the sampling instant *t_n_* = *nT_s_*, the terahertz wave signal emitted by the transmitter can be described as(1)$$S_{t}\left(t_{n}\right)\;=\;Ae^{j\omega t}\;=\;Ae^{j2\pi f_{0}t_{n}},$$
where *A* is the amplitude of the signal, and *ω* = 2π*f*_0_ is the frequency of the terahertz wave signal. Coded aperture A is composed of *G_A_* coding elements, and thus the signal at its *a*-th coding element is(2)$$S_{a}\left(t_{n}\right)\;=\;Ae^{j\left(2\pi f_{0}t_{n}\;-\;\frac{\omega }{c}d_{Ta}\right)}\;=\;Ae^{j2\pi f_{0}\left(t_{n}\;-\;\frac{d_{Ta}}{c}\right)}\;=\;Ae^{j2\pi f_{0}\left(t_{n}\;-\;t_{Ta}\right)},$$
where c is the light speed, *d_Ta_* is the distance between the transmitter and the *a*-th coding element of coded aperture A, and *t_Ta_* = *d_Ta_*/c is its corresponding time delay. Through random discrete phase coding, the signal transmitted through coded aperture A is converted into(3)$$S_{Am}(t_{n})\;=\;A\sum_{a\;=\;1}^{G_{A}}e^{j2\pi f_{0}\left(t_{n}\;-\;t_{Ta}\right)}\;e^{j\varphi_{a}\left(t_{n}\right)},$$
where *φ_a_*(*t_n_*) represents the coding factor of *a*-th coding element at the sampling instant *t_n_*. Similarly, the scattered signal reflected from the *k*-th scatterer of the imaging plane at the sampling instant *t_n_* is(4)$$S_{k}\left(t\;=\;t_{n}\right)\;=\;A\sigma_{k}^{\prime }\sum_{a\;=\;1}^{G_{A}}\left[e^{j2\pi f_{0}\left(t_{n}-t_{Ta}-t_{ak}\right)}\;e^{j\varphi_{a}\left(t_{n}\right)}\right],$$
and that modulated by the *b*-th coding element of coded aperture B is(5)$$S_{Bm}\left(t_{n}\right)\;=\;Ae^{j\varphi_{b}\left(t_{n}\right)}\sum_{k\;=\;1}^{K}\left[\sigma_{k}^{\prime }\sum_{a\;=\;1}^{G_{A}}\left(e^{j2\pi f_{0}\left(t_{n}-t_{Ta}-t_{ak}-t_{kb}\right)}\;e^{j\varphi_{a}\left(t_{n}\right)}\right)\right],$$
respectively. In Equations (4) and (5), $\sigma_{k}^{\prime }$
is the scattering coefficient corresponding to the *k*-th scatterer of the imaging plane; the phase shift *φ_b_*(*t_n_*) is the coding factor of *b*-th coding element of coded aperture B; *d_ak_* is the distance between the *a*-th coding element of coded aperture A and the *k*-th scatterer; *d_kb_* is the distance between the *k*-th scatterer and the *b*-th coding element of coded aperture B; *t_ak_* = *d_ak_*/c; and *t_kb_* = *d_kb_*/c. Therefore, the echo signal captured by the receiver at the sampling instant *t_n_* can be expressed as(6)$$S_{rn}\left(t_{n}\right)\;=\;A\sum_{b\;=\;1}^{G_{B}}\left\{e^{j\varphi_{b}\left(t_{n}\right)}\sum_{k\;=\;1}^{K}\left[\sigma_{k}^{\prime }\sum_{a\;=\;1}^{G_{A}}\left(e^{j2\pi f_{0}\left(t_{n}-t_{Ta}-t_{ak}-t_{kb}-t_{bR}\right)}\;e^{j\varphi_{a}\left(t_{n}\right)}\right)\right]\right\}\;=\;\sum_{k\;=\;1}^{K}\sigma_{k}^{\prime }S_{t}^{\prime }\left(t_{n},k\right),$$
where *G_B_* is the number of coding elements of coded aperture B; *d_bR_* is the distance between the *b*-th coding element of coded aperture B and the receiver; *t_bR_* = *d_bR_*/c; and(7)$$S_{t}^{\prime }\left(t_{n},k\right)\;=\;A\sum_{b\;=\;1}^{G_{B}}\left\{\sum_{a\;=\;1}^{G_{A}}e^{j2\pi f_{0}\left(t_{n}-t_{Ta}-t_{ak}-t_{kb}-t_{bR}\right)}\;e^{j\left[\varphi_{a}\left(t_{n}\right)+\varphi_{b}\left(t_{n}\right)\right]}\right\},$$
is the reference signal of the *k*-th grid point of the imaging plane at the sampling instant *t_n_*. Following *N* sampling iterations of the scattered wave field, the received echo signals can be compactly represented in matrix form as(8)$$\begin{array}{l} S_{r}\;=\;S_{t}\cdot \sigma^{\prime }+w\\ \left[\begin{array}{c} S_{r1}\left(t_{1}\right)\\ S_{r2}\left(t_{2}\right)\\ S_{r3}\left(t_{3}\right)\\ \vdots \\ S_{rN}\left(t_{N}\right) \end{array}\right]\;=\;\left[\begin{array}{ccccc} S_{t}^{\prime }\left(t_{1},1\right) & S_{t}^{\prime }\left(t_{1},2\right) & S_{t}^{\prime }\left(t_{1},3\right) & \ldots & S_{t}^{\prime }\left(t_{1},K\right)\\ S_{t}^{\prime }\left(t_{2},1\right) & S_{t}^{\prime }\left(t_{2},2\right) & S_{t}^{\prime }\left(t_{2},3\right) & \ldots & S_{t}^{\prime }\left(t_{2},K\right)\\ S_{t}^{\prime }\left(t_{3},1\right) & S_{t}^{\prime }\left(t_{3},2\right) & S_{t}^{\prime }\left(t_{3},3\right) & \ldots & S_{t}^{\prime }\left(t_{3},K\right)\\ \vdots & \vdots & \vdots & \ddots & \vdots \\ S_{t}^{\prime }\left(t_{N},1\right) & S_{t}^{\prime }\left(t_{N},2\right) & S_{t}^{\prime }\left(t_{N},3\right) & \ldots & S_{t}^{\prime }\left(t_{N},K\right) \end{array}\right]\left[\begin{array}{c} \sigma_{1}^{\prime }\\ \sigma_{2}^{\prime }\\ \sigma_{3}^{\prime }\\ \vdots \\ {\sigma^{\prime }}_{K} \end{array}\right]+\left[\begin{array}{c} w\left(t_{1}\right)\\ w\left(t_{2}\right)\\ w\left(t_{3}\right)\\ \vdots \\ w\left(t_{N}\right) \end{array}\right] \end{array},$$
where ***S_r_***∈*C^N^*^×1^ is the echo vector, ***S_t_***∈*C^N^*^×*K*^ is the RSM comprising the elements per Equation (7), ***σ*’** = [*σ*′_1_, *σ*′_2_, *σ*′_3_, …, *σ*′*_K_*]^T^ is the scattering-coefficient vector with each element corresponding to each scatterer of the imaging plane, and ***w***∈*C^N^*^×1^ is the noise vector. Target reconstruction essentially involves solving the non-homogeneous linear system per Equation (8) to determine the scattering-coefficient vector ***σ*’**.

### 2.2. Estimation Method of Target Reconstruction Imaging

As shown in Equation (8), the accuracy of the scattering-coefficient vector critical for imaging quality heavily relies on the mathematical properties of RSM ***S_t_***, such as rank-related characteristics. Per Equation (7), RSM ***S_t_*** embeds the spatiotemporal information in its columns and rows: each column ***X_k_*** = [*S′_t_*(*t*_1_,*k*), *S′_t_*(*t*_2_,*k*), *S′_t_*(*t*_3_,*k*), ···, *S′_t_*(*t_N_*,*k*)]^T^ with *k* = 1, 2, 3, ···, *K* corresponds to the *k*-th scatterer across all *N* sampling instants, while each row ***Y_n_*** = [*S′_t_*(*t_n_*,1), *S′_t_*(*t_n_*,2), *S′_t_*(*t_n_*,3), ···, *S′_t_*(*t_n_*,*K*)] with *n* = 1, 2, 3, ···, *N* corresponds to the *n*-th sampling instant across all *K* scatters on the imaging plane. Thus, RSM ***S_t_*** can be expressed as(9)$$S_{t}\;=\;\left[\begin{array}{ccccccc} X_{1} & X_{2} & X_{3} & \ldots & X_{k} & \ldots & X_{K} \end{array}\right]\;=\;{\left[\begin{array}{ccccccc} Y_{1} & Y_{2} & Y_{3} & \ldots & Y_{n} & \ldots & Y_{N} \end{array}\right]}^{T},$$

Under ideal conditions, the column vectors ***X_k_*** of RSM ***S_t_*** are mutually uncorrelated, as are row vectors ***Y_n_***, establishing optimal spatiotemporal diversity for the entire imaging process. Thus, the spatial and temporal independence functions defined by the mean column and row correlations as(10)$$\gamma_{\mathrm{space}}\;=\;\frac{2\sum_{k_{1}\;=\;1}^{K-1}\left[\sum_{k_{2}\;=\;k_{1}+1}^{K}\rho \left(X_{k_{1}},X_{k_{2}}\right)\right]}{K\left(K-1\right)},$$(11)$$\gamma_{\mathrm{time}}\;=\;\frac{2\sum_{n_{1}\;=\;1}^{N-1}\left[\sum_{n_{2}\;=\;n_{1}+1}^{N}\rho \left(Y_{n_{1}},Y_{n_{2}}\right)\right]}{N\left(N-1\right)},$$
can be used to estimate the imaging performance. In Equations (10) and (11),(12)$$\rho \left(X_{k_{1}},X_{k_{2}}\right)\;=\;\frac{\sum_{n\;=\;1}^{N}\left[S_{t}^{\prime }\left(t_{n},k_{1}\right)-{\bar{X}}_{k_{1}}\right]\left[S_{t}^{\prime }\left(t_{n},k_{2}\right)-{\bar{X}}_{k_{2}}\right]}{\sqrt{\sum_{n\;=\;1}^{N}{\left[S_{t}^{\prime }\left(t_{n},k_{1}\right)-{\bar{X}}_{k_{1}}\right]}^{2}}\sqrt{\sum_{n\;=\;1}^{N}{\left[S_{t}^{\prime }\left(t_{n},k_{2}\right)-{\bar{X}}_{k_{2}}\right]}^{2}}},$$(13)$$\rho \left(Y_{n_{1}},Y_{n_{2}}\right)\;=\;\frac{\sum_{k\;=\;1}^{K}\left[S_{t}^{\prime }\left(t_{n_{1}},k\right)-{\bar{Y}}_{n_{1}}\right]\left[S_{t}^{\prime }\left(t_{n_{2}},k\right)-{\bar{Y}}_{n_{2}}\right]}{\sqrt{\sum_{k\;=\;1}^{K}{\left[S_{t}^{\prime }\left(t_{n_{1}},k\right)-{\bar{Y}}_{n_{1}}\right]}^{2}}\sqrt{\sum_{k\;=\;1}^{K}{\left[S_{t}^{\prime }\left(t_{n_{2}},k\right)-{\bar{Y}}_{n_{2}}\right]}^{2}}},$$
are the column and row correlation coefficients, where ${\bar{X}}_{k}\;=\;\sum_{n\;=\;1}^{N}S_{t}^{\prime }\left(t_{n},k\right)/N$, and ${\bar{Y}}_{n}\;=\;\sum_{k\;=\;1}^{K}S_{t}^{\prime }\left(t_{n},k\right)/K$, *k* = 1, 2, 3, ···, *K*, and *n* = 1, 2, 3, ···, *N*. The imaging quality increases monotonically as *γ*_space_ and *γ*_time_ decrease.

Moreover, another quantitative method to evaluate the resolving ability by analyzing RSM ***S_t_*** depends on the effective rank, which is a measure of the number of linearly independent components of RSM ***S_t_*** [27]. The effective rank *n_r_* is determined as the smallest integer satisfying the inequality(14)$$\frac{\sqrt{\sum_{i\;=\;1}^{n_{r}}\delta_{i}^{2}}}{\sqrt{\sum_{i\;=\;1}^{n_{R}}\delta_{i}^{2}}}\;=\;\frac{\sqrt{\delta_{1}^{2}+\delta_{2}^{2}+\delta_{3}^{2}+\ldots +\delta_{n_{r}}^{2}}}{\sqrt{\delta_{1}^{2}+\delta_{2}^{2}+\delta_{3}^{2}+\ldots +\delta_{n_{r}}^{2}+\ldots +\delta_{n_{R}}^{2}}}\geq \xi ,$$
where $n_{R}$ is the rank of the RSM ***S_t_***, *δ*_1_ ≥ *δ*_2_ ≥ … ≥ $\delta_{n_{r}}$ ≥ … ≥ $\delta_{n_{R}}$ > 0 are the singular values sorted in descending order, and *ξ* < 1 but is close to 1. The effective rank *n_r_* quantifies the dominant singular values determining RSM ***S_t_***’s reconstruction capability.

After target reconstruction is achieved by solving Equation (8), the resultant imaging fidelity can be rigorously quantified by the probability of successful imaging (PSI) and the relative imaging error (RIE), which are defined as(15)$$\mathrm{PSI}\;=\;\frac{\min{\left(\sigma_{x}^{\prime }\right)}_{\Omega }}{\max{\left({\bar{\sigma }}_{x}^{\prime }\right)}_{\Omega }},$$(16)$$\mathrm{RIE}\;=\;20\mathrm{lg}\frac{{\left\|\sigma_{x}^{\prime }-\sigma^{\prime }\right\|}_{2}}{{\left\|\sigma^{\prime }\right\|}_{2}},$$
where ***σ*′*_x_*** and ***σ’*** are the calculated and actual scattering-coefficient vectors, respectively, and ${\left\|\cdot \right\|}_{2}$ denotes the *L*_2_-norm. The subscript Ω denotes the set of the indices corresponding to the dominant scatterers with non-zero actual scattering coefficient. The values of (***σ*′*_x_***)_Ω_ within Ω are the same as those of the calculated scattering-coefficient vector ***σ*′*_x_***, while ${\left({\bar{\sigma }}_{x}^{\prime }\right)}_{T}$ is defined as(17)$${\left({\bar{\sigma }}_{xk}^{\prime }\right)}_{T}\;=\;\left\{\begin{array}{cc} 0 & k\in \Omega \\ \sigma_{xk}^{\prime } & k\notin \Omega \end{array}\right.,$$

PSI quantifies the separation between reconstructed scattering components and noise, with higher values indicating greater confidence in target identification. Conversely, the RIE measures the ratio of the reconstruction error to the true scattering coefficients. Lower RIE values signify higher reconstruction fidelity.

Equation (8) reveals that RSM **S*_t_*** structurally encodes the imaging physics, where columns contain reference signals for individual scatterers across all *N* sampling instants, while rows comprise full-scene snapshots at single instants. Consequently, spatiotemporal field independence manifests mathematically as row/column decorrelation in RSM **S*_t_***. This decorrelation degree, which is quantified by the effective rank, directly determines azimuth–elevation resolution: enhanced decorrelation coupled with a higher effective rank enables superior resolution. Phase interactions between coding elements and target scatterers fundamentally govern RSM **S*_t_***’s correlation properties. Given the mathematical equivalence between the transmitting-side and receiving-side random discrete phase coding [25], this work focuses on phase interactions originating from coding elements of the coded aperture at the transmitting terminal to scatterers(18)$$\phi_{0}\left(a,k\right)\;=\;-2\pi f_{0}\left(t_{Ta}+t_{ak}\right),$$
since it dictates RSM ***S_t_***’s spatiotemporal independence characteristics, which are essential for target reconstruction imaging.

## 3. Effect of Imaging Range on Column and Row Correlation of RSM

This section investigates the effect of the imaging range on the spatiotemporal independence characteristics of RSM ***S_t_*** through theoretical and quantitative analysis, with primary parameters listed in Table 1. The frequency is chosen to be 670 GHz, which is a technologically feasible high-frequency point within an atmospheric window to enhance imaging resolution [28,29]. A signal-to-noise ratio (SNR) of 20 dB, a typical value in terahertz imaging studies, was set to simulate a practical scenario [9,25,26,30]. Identical coded apertures A and B, as shown in Figure 2, employ $a_{C}\times b_{C}\;=\;2\;\mathrm{cm}\times 2\;\mathrm{cm}$ axial inter-element pitch to reduce system complexity while implementing a 1-bit random discrete phase-encoding scheme of ±0.5π phase shifts that delivers optimal resolution performance [26]. The imaging plane of the size *A_E_* × *B_E_* = 22 mm × 22 mm is discretized by *M_E_* vertical and *N_E_* horizontal grid lines, forming (*M_E_* − 1) × (*N_E_* − 1) rectangular cells with *K* = *M_E_* × *N_E_* = 2025 scatterers located exclusively at the cell vertices, as shown in Figure 3, comprising 16 dominant scatterers highlighted in red with the scattering coefficient *σ*′ = 1 representing four types of resolution—1 cm, 2 cm, 3 cm, and 4 cm—alongside 2009 dummy scatterers with *σ*′ = 0.

### 3.1. Research on Column Correlation of RSM

In the linear imaging matrix Equation (8), the columns of the RSM ***S_t_*** constitute the spatial measurement basis. Each column corresponds to a specific scatterer on the imaging plane, encoding its modulation characteristics under terahertz probing signals and serving as the dictionary for solving the target scattering coefficient *σ*′. Consequently, RSM column correlation reflects spatial independence, and thus is a critical determinant of imaging resolution and scattering coefficient recovery accuracy.

#### 3.1.1. Theoretical Analysis

In RSM ***S_t_***, defined in Equations (7) and (8), the pairwise correlation between any *k*_1_-th and *k*_2_-th columns (1 ≤ *k*_1_, *k*_2_ ≤ *K*) quantifies the spatial correlation between their corresponding scatterers. This correlation arises only from differences in the distinct entries $t_{ak_{1}}$, $t_{k_{1}b}$ and $t_{ak_{2}}$, $t_{k_{2}b}$, which govern the relative phase relationships in wave propagation from each coding element to the corresponding *k*_1_-th and *k*_2_-th scatterer. Consequently, these coordinate differences primarily govern RSM column correlation. Due to mathematical equivalence between transmit and receive coding schemes, this paper focuses on the transmit-side phase difference vector $\phi_{a}\left(k_{1},k_{2}\right)\in R^{G_{A}\times 1}$ between wavefronts propagating from each coding element at the transmitting terminal to the *k*_1_-th and *k*_2_-th scatterers, defined as(19)$$\phi_{a}\left(k_{1},k_{2}\right)\;=\;\phi_{0}\left(a,k_{2}\right)-\phi_{0}\left(a,k_{1}\right)\;=\;\frac{2\pi f_{0}}{c}\left\{\sqrt{{\left(x_{a}-x_{k_{1}}\right)}^{2}+{\left(y_{a}-y_{k_{1}}\right)}^{2}+L^{2}}-\sqrt{{\left(x_{a}-x_{k_{2}}\right)}^{2}+{\left(y_{a}-y_{k_{2}}\right)}^{2}+L^{2}}\right\},$$
where *L* is the imaging range; (***x_a_***, ***y_a_***) are the Cartesian coordinates of the *a*-th coding element on the coded aperture, as shown in Figure 2; $\left(x_{k_{1}},y_{k_{1}}\right)$ and $\left(x_{k_{2}},y_{k_{2}}\right)$ are those of the *k*_1_-th and *k*_2_-th scatterers on the imaging plane, as shown in Figure 3, respectively, with 1 ≤ *a* ≤ *G_A_* = *M_C_* × *N_C_* for the coded aperture and 1 ≤ *k*_1_, *k*_2_ ≤ *M_E_* × *N_E_* for the imaging plane.

If the value of the phase difference vector $\phi_{a}\left(k_{1},k_{2}\right)$ per Equation (19) exhibits a narrow concentration within a specific numerical range, the corresponding *k*_1_-th and *k*_2_-th columns in RSM become strongly correlated. Specifically, further analysis demonstrates that for any pair of adjacent (both row-wise and column-wise) *a*_1_-th and *a*_2_-th coding elements at the transmitting terminal, if the relative difference between $\phi_{a_{1}}\left(k_{1},k_{2}\right)$ and $\phi_{a_{2}}\left(k_{1},k_{2}\right)$(20)$$\Delta \phi_{a}\left(k_{1},k_{2}\right)\;=\;\phi_{a_{2}}\left(k_{1},k_{2}\right)-\phi_{a_{1}}\left(k_{1},k_{2}\right),$$
equals 2*m*′π where *m*′ = 0, 1, 2, …, the value of the phase difference vector $\phi_{a}\left(k_{1},k_{2}\right)$ for each coding element becomes highly concentrated within a specific numerical range. Consequently, the *k*_1_-th and *k*_2_-th columns in RSM ***S_t_*** exhibit a strong correlation.

#### 3.1.2. Mathematical Derivation

Under the far-field condition *L* >> |*x_a_* − *x_k_*| and *L* >> |*y_a_* − *y_k_*|, the relative difference $\Delta \phi_{a}\left(k_{1},k_{2}\right)$ per Equation (20) is approximated as(21)$$\Delta \phi_{a}\left(k_{1},k_{2}\right)\;=\;\phi_{a_{2}}\left(k_{1},k_{2}\right)-\phi_{a_{1}}\left(k_{1},k_{2}\right)\approx \frac{2\pi f_{0}}{Lc}\left[\left(x_{a_{2}}-x_{a_{1}}\right)\left(x_{k_{2}}-x_{k_{1}}\right)+\left(y_{a_{2}}-y_{a_{1}}\right)\left(y_{k_{2}}-y_{k_{1}}\right)\right],$$
where $\left(x_{a_{1}},y_{a_{1}}\right)$ and $\left(x_{a_{2}},y_{a_{2}}\right)$ denote coordinates of the *a*_1_-th and *a*_2_-th coding element on the coded aperture, as shown in Figure 2. Δ*k_M_* and Δ*k_N_* denote the row and column index differences between the *k*_1_-th and *k*_2_-th scatterers, respectively. Thus $x_{k_{2}}-x_{k_{1}}\;=\;-\Delta k_{M}\cdot b_{E}$ and $y_{k_{2}}-y_{k_{1}}\;=\;\Delta k_{N}\cdot a_{E}$.

When the adjacent *a*_1_-th and *a*_2_-th coding elements of the coded aperture share the same row, if(22)$$\Delta \phi_{a}\left(k_{1},k_{2}\right)\approx \frac{2\pi f_{0}}{Lc}\left(y_{a_{2}}-y_{a_{1}}\right)\left(y_{k_{2}}-y_{k_{1}}\right)\;=\;\frac{2\pi f_{0}}{Lc}a_{C}\cdot \Delta k_{N}a_{E}\;=\;2m^{\prime }\pi ,$$
where *m*′ = 0, 1, 2, …, the column index difference between the *k*_1_-th and *k*_2_-th scatterers on the imaging plane must be $\Delta k_{N}\;=\;m^{\prime }Lc/f_{0}a_{C}a_{E}$ and $Lc/f_{0}a_{C}a_{E}$ must be an integer. And when the adjacent *a*_1_-th and *a*_2_-th coding elements of the coded aperture share the same column, if(23)$$\Delta \phi_{a}\left(k_{1},k_{2}\right)\approx \frac{2\pi f_{0}}{Lc}\left(x_{a_{2}}-x_{a_{1}}\right)\left(x_{k_{2}}-x_{k_{1}}\right)\;=\;\frac{2\pi f_{0}}{Lc}b_{C}\cdot \Delta k_{M}b_{E}\;=\;2m^{\prime }\pi ,$$
where *m*′ = 0, 1, 2, …, the row index difference between the *k*_1_-th and *k*_2_-th scatterers on the imaging plane must be $\Delta k_{M}\;=\;m^{\prime }Lc/f_{0}b_{C}b_{E}$ and $Lc/f_{0}b_{C}b_{E}$ must be an integer.

For typical symmetric systems where $a_{C}\;=\;b_{C}$ and $a_{E}\;=\;b_{E}$ as shown in Table 1, $Lc/f_{0}a_{C}a_{E}$ being an integer serves as a sufficient criterion to assess column correlation. The preceding analysis indicates that if $Lc/f_{0}a_{C}a_{E}$ is precisely an integer, for the *k*_1_-th and *k*_2_-th scatterers whose column and row index differences are an integer (including 0) multiple of $Lc/f_{0}a_{C}a_{E}$, the phase difference vector $\phi_{a}\left(k_{1},k_{2}\right)$ per Equation (19) exhibits a narrow concentration within a specific numerical range.

For a representative case with the imaging range *L* = 6.7 m, the criterion $Lc/f_{0}a_{C}a_{E}\;=\;30$ is an integer, indicating the existence of strongly correlated columns in the RSM ***S_t_***. As shown in Figure 4, column 277 of RSM ***S_t_*** exhibits near-unity correlation with columns 307, 1627, and 1657. This stems from the narrow concentration of the phase difference vector $\phi_{a}\left(k_{1}\;=\;277,k_{2}\right)$, as shown in Figure 5. In Figure 5, $\phi_{a}\left(277,307\right)$ remains stable at ~1.5π across all coding elements of the coded aperture, while $\phi_{a}\left(277,1627\right)$ and $\phi_{a}\left(277,1657\right)$ concentrate at ~1.5π and ~π, respectively. Such a phase concentration arises from the relative coordinate positions of these four scatterers on the imaging plane as detailed in Table 2, where their row and column index differences are exclusively either 0 or 30, determined by ${\left.Lc/f_{0}a_{C}a_{E}\right|}_{L\;=\;6.7\;m}$. These differences govern the correlation structure of RSM ***S_t_*** as visualized in Figure 6.

For a counterexample with the imaging range *L* = 2.5 m, the criterion $Lc/f_{0}a_{C}a_{E}\;=\;11.19$ implies that columns of RSM show considerable correlation when their corresponding scatterers with the row and column index differences Δ*k_M_* and Δ*k_N_* on the imaging plane are both around 11.19. Column 1365 is considerably correlated with eight columns of RSM ***S_t_*** as shown in Figure 7, and their corresponding scatterers located on the imaging plane with row and column index differences of Δ*k_M_* and Δ*k_N_* = −11, 0 and 11 determined by ${\left.Lc/f_{0}a_{C}a_{E}\right|}_{L\;=\;2.5\;m}$, as shown in Table 3. However, the non-integer criterion ${\left.Lc/f_{0}a_{C}a_{E}\right|}_{L\;=\;2.5\;m}$ suppresses peak correlations below 0.4, as shown in Figure 7. This suppression stems from phase difference vectors $\phi_{a}\left(k_{1},k_{2}\right)$ as shown in Figure 8, which span a broad portion of the [0, 2π] range for the four strongest correlations, contrasting sharply with the narrow concentrations observed at *L* = 6.7 m, as demonstrated in Figure 5. Such phase difference diversity prevents the column correlations from approaching unity. Consequently, the overall column correlation at *L* = 2.5 m as shown in Figure 9 is markedly lower than that with *L* = 6.7 m, as shown in Figure 6, which is directly attributable to the non-integer versus integer nature of the respective criteria $Lc/f_{0}a_{C}a_{E}$.

### 3.2. Research on Row Correlation of RSM

The row correlation of RSM ***S_t_*** characterizes the temporal independence of the terahertz radiation field across different sampling instants, which directly quantifies the correlation among the equations as per Equation (6), constituting the linear imaging system of Equation (8). Reduced inter-equation correlation enables well-posed inversion for accurate reconstruction of the target’s scattering coefficient vector ***σ***′ = [*σ*′_1_, *σ*′_2_, *σ*′_3_, …, *σ*′*_K_*]^T^.

#### 3.2.1. Theoretical Analysis

In RSM ***S_t_*** per Equation (8), its entries per Equation (7) differ primarily through the phase term 2π*f*_0_*t_n_* + *φ_a_*(*t_n_*) + *φ_b_*(*t_n_*). At each sampling instant *t_n_*, the carrier term $e^{j2\pi f_{0}t_{n}}\;=\;1$ renders the contribution of 2π*f*_0_*t_n_* negligible. The composite factor *φ_a_*(*t_n_*) + *φ_b_*(*t_n_*) represents discrete random coding applied by coding elements to introduce temporal diversity into terahertz signals at sampling instant *t_n_*. This illuminates scatterers with distinct radiation patterns across all sampling instants. When these discretely modulated wavefronts interact with scatterers, echoes acquire unique signatures enabling precise target discrimination during reconstruction. As a result, variations in the path length-induced phase difference among different coding elements of the coded aperture for each scatterer on the imaging plane determine the coding efficacy of the random discrete phase shift *φ_a_*(*t_n_*) + *φ_b_*(*t_n_*). Such phase diversity generates temporal radiation patterns that are essential for target resolution in the linear system per Equation (8). Owing to mathematical equivalence between transmit/receive coding schemes, this paper focuses on analyzing the path length-induced phase difference vector $\phi_{k}\left(a_{1},a_{2}\right)\in R^{1\times K}$ from all adjacent (both row-wise and column-wise) *a*_1_- and *a*_2_-th coding elements at the transmitting terminal to each scatterer:(24)$$\phi_{k}\left(a_{1},a_{2}\right)\;=\;\phi_{0}\left(a_{2},k\right)-\phi_{0}\left(a_{1},k\right)\;=\;2\pi f_{0}\left(t_{Ta_{1}}-t_{Ta_{2}}\right)+\frac{2\pi f_{0}}{c}\left\{\sqrt{{\left(x_{a_{1}}-x_{k}\right)}^{2}+{\left(y_{a_{1}}-y_{k}\right)}^{2}+L^{2}}-\sqrt{{\left(x_{a_{2}}-x_{k}\right)}^{2}+{\left(y_{a_{2}}-y_{k}\right)}^{2}+L^{2}}\right\},$$

If the path length-induced phase difference vector $\phi_{k}\left(a_{1},a_{2}\right)$ per Equation (24) clusters around sparsely spaced, highly discrete specific values rather than spanning the full [0, 2π] range, the random phase-encoding efficacy degrades severely. Consequently, temporal independence quantified by the row correlation of RSM ***S_t_*** deteriorates. Furthermore, for any pair of adjacent (both row-wise and column-wise) *k*_1_-th and *k*_2_-th scatterers on the imaging plane, if the relative difference between $\phi_{k_{1}}\left(a_{1},a_{2}\right)$ and $\phi_{k_{2}}\left(a_{1},a_{2}\right)$(25)$$\Delta \phi_{k}\left(a_{1},a_{2}\right)\;=\;\phi_{k_{2}}\left(a_{1},a_{2}\right)-\phi_{k_{1}}\left(a_{1},a_{2}\right),$$
equals $2\pi /n^{\prime }$ where *n*′ = 0, 1, 2, …, the path length-induced phase difference vector $\phi_{k}\left(a_{1},a_{2}\right)$ for all the scatterers will cluster around a set of highly discrete specific values spaced at interval $\Delta \phi_{k}\left(a_{1},a_{2}\right)$ per Equation (25). This clustering impairs the effectiveness of random discrete phase-encoding, thereby compromising the temporal independence of RSM ***S_t_***.

#### 3.2.2. Mathematical Derivation

Under the far-field condition *L* >> |*x_a_* − *x_k_*| and *L* >> |*y_a_* − *y_k_*|, the relative difference $\Delta \phi_{k}\left(a_{1},a_{2}\right)$ per Equation (25) is approximated as(26)$$\Delta \phi_{k}\left(a_{1},a_{2}\right)\;=\;\phi_{k_{2}}\left(a_{1},a_{2}\right)-\phi_{k_{1}}\left(a_{1},a_{2}\right)\approx \frac{2\pi f_{0}}{Lc}\left[\left(x_{a_{2}}-x_{a_{1}}\right)\left(x_{k_{2}}-x_{k_{1}}\right)+\left(y_{a_{2}}-y_{a_{1}}\right)\left(y_{k_{2}}-y_{k_{1}}\right)\right],$$

For the adjacent *a*_1_-th and *a*_2_-th coding elements, their relative coordinate positions are $x_{a_{2}}-x_{a_{1}}\;=\;-b_{C}$, $y_{a_{1}}\;=\;y_{a_{2}}$ or $x_{a_{1}}\;=\;x_{a_{2}}$, $y_{a_{2}}-y_{a_{1}}\;=\;a_{C}$.

When the adjacent *k*_1_-th and *k*_2_-th scatterer on the imaging plane are in the same row, if(27)$$\Delta \phi_{k}\left(a_{1},a_{2}\right)\approx \frac{2\pi f_{0}}{Lc}\left(y_{a_{2}}-y_{a_{1}}\right)\left(y_{k_{2}}-y_{k_{1}}\right)\;=\;2\pi \frac{f_{0}a_{C}a_{E}}{Lc}\;=\;\frac{2\pi }{n^{\prime }},$$
where *n*′ = 0, 1, 2, …, $Lc/f_{0}a_{C}a_{E}\;=\;n^{\prime }$ must be an integer. And when the adjacent *k*_1_-th and *k*_2_-th scatterer on the imaging plane in the same column, if(28)$$\Delta \phi_{k}\left(a_{1},a_{2}\right)\approx \frac{2\pi f_{0}}{Lc}\left(x_{a_{2}}-x_{a_{1}}\right)\left(x_{k_{2}}-x_{k_{1}}\right)\;=\;2\pi \frac{f_{0}b_{C}b_{E}}{Lc}\;=\;\frac{2\pi }{n_{M}},$$
where *n*′ = 0, 1, 2, …, $Lc/f_{0}b_{C}b_{E}\;=\;n^{\prime }$ must be an integer.

When Equations (27) and (28) are fulfilled, the path length-induced phase difference vector $\phi_{k}\left(a_{1},a_{2}\right)$ per Equation (24) clusters around a discrete set of values with $\Delta \phi_{k}\left(a_{1},a_{2}\right)$ spacing.

Consider imaging range *L* = 2.2333 m, where the criterion $Lc/f_{0}a_{C}a_{E}\;=\;10$ is an integer. For any pair of adjacent *k*_1_-th and *k*_2_-th scatterers on the imaging plane, the relative difference per Equation (25) $\Delta \phi_{k}\left(a_{1},a_{2}\right)\;=\;0.6283\;\mathrm{rad}\;=\;2\pi /10$. Consequently, the path length-induced phase difference vector is $\phi_{k}\left(a_{1},a_{2}\right)$ per Equation (24) for all scatterers on the imaging plane cluster near a set of highly discrete specific values spaced at interval $\Delta \phi_{k}\left(a_{1},a_{2}\right)\;=\;2\pi /10$, as shown in Figure 10a. Conversely, at *L* = 5.0 m, where the criterion $Lc/f_{0}a_{C}a_{E}\;=\;22.3881$ is not an integer, $\phi_{k}\left(a_{1},a_{2}\right)$ uniformly spans [0, 2*π*], which is nearly the entire [0, 2π] range, as evidenced in Figure 10b. This confirms the theoretical principle: clustered phases enhance the row correlation of RSM ***S_t_***, while the diverse distributions suppress it, as shown in Figure 11a,b.

### 3.3. Summary of Unified Physical Insights

The preceding theoretical and quantitative analyses reveal that the degradation of both column and row correlations of RSM shares a common physical root, as shown in Figure 12. As the wavelength is $\lambda \;=\;c/f_{0}$, the dimensionless criterion can be rewritten as $Lc/f_{0}a_{C}a_{E}\;=\;L/\left(a_{C}a_{E}/\lambda \right)$. When $Lc/f_{0}a_{C}a_{E}\;=\;L/\left(a_{C}a_{E}/\lambda \right)$ is precisely an integer, the path length difference from any adjacent coding elements to some specific scatterer pairs becomes an integer multiple of λ, which triggers a dual degradation mechanism.

First, for any scatterer pairs with the row and column index differences of integer multiples (including 0) of $Lc/f_{0}a_{C}a_{E}$, the phase difference vector $\phi_{a_{1}}\left(k_{1},k_{2}\right)$ in Figure 12 from the entire coded aperture concentrates within a specific numerical range, as shown in Figure 5. The system is thus unable to distinguish between *k*_1_- and *k*_2_-th scatterers, since they present identical phase modulation histories throughout the sampling process. This results in strong correlation between their corresponding columns in RSM.

Second, the same integer $Lc/f_{0}a_{C}a_{E}$ causes the path length-induced phase difference vector $\phi_{k_{1}}\left(a_{1},a_{2}\right)\;=\;\phi_{0}\left(a_{2},k_{1}\right)-\phi_{0}\left(a_{1},k_{1}\right)$ in Figure 12 from all adjacent coding element pairs to each scatterer to cluster around a sparse set of discrete values as shown in Figure 10a rather than uniformly covering [0, 2π]. This structured, discrete phase distribution dramatically diminishes the randomness that the applied discrete phase shift *φ_a_*(*t_n_*) + *φ_b_*(*t_n_*) can impart to the radiation pattern. Instead of producing unique patterns at each sampling instant, the coding process yields only a limited set of predictable, periodic patterns. This loss of temporal diversity manifests as strong correlation among different rows of RSM, reducing the independence of the equations in the linear system.

The analysis reveals analogous mathematical forms between column correlation criteria per Equations (19) and (20) and row correlation criteria per Equations (24) and (25). This similarity indicates strong coupling between column and row correlations of RSM ***S_t_***. Crucially, whether $Lc/f_{0}a_{C}a_{E}$ is an integer is the unifying determinant for both spatial and temporal independence.

## 4. Numerical Simulation Research

The above analysis demonstrates that spatiotemporal independence quantified by column and row correlations of RSM ***S_t_*** can be assessed by the criteria $Lc/f_{0}a_{C}a_{E}$ rather than being inversely proportional to the imaging range *L*. Consequently, TCAI quality varies non-monotonically with the imaging range *L*. These findings are investigated in this section by analyzing the range-dependent PSI and RIE of the imaging results collaborated with evaluating the spatiotemporal independence and the effective rank.

### 4.1. Target Reconstruction Imaging Algorithm

The sparse target is reconstructed by solving the TCAI linear imaging system per Equation (8) using the Two-step Iterative Shrinkage/Thresholding (TwIST) algorithm [31]. TwIST solves the optimization problem to iteratively minimize the composite objective function:(29)$$f\left(\sigma^{\prime }\right)\;=\;\underset{\sigma^{\prime }}{\min}0.5{\left\|S_{r}-S_{t}\sigma^{\prime }\right\|}_{2}^{2}+\lambda {\left\|\sigma^{\prime }\right\|}_{1},$$
consisting of the data fidelity term and the regularization term promoting sparsity. In Equation (29), *λ* is the regularization parameter which balances fidelity and sparsity, and ${\left\|\cdot \right\|}_{1}$ denotes the *L*_1_-norm. Each iteration performs a forward gradient step:(30)$$\sigma_{x2}^{\prime }\;=\;\sigma_{x1}^{\prime }+\frac{S_{t}^{H}\cdot \left(S_{r}-S_{t}\cdot \sigma_{x1}^{\prime }\right)}{L},$$
followed by a soft-thresholding operation:(31)$$\sigma_{x3}^{\prime }\;=\;S_{\lambda /L}\left(\sigma_{x2}^{\prime }\right),$$
where $S_{t}^{H}$ is the Hermitian transpose of ***S_t_***, $L\;=\;{\left\|S_{t}^{H}\cdot S_{t}\right\|}_{2}$ is the Lipschitz constant, and $S_{\lambda /L}\left(\sigma_{x2}^{\prime }\right)$ is the soft-thresholding operator:(32)$$S_{\lambda /L}\left(\sigma_{x2}^{\prime }\right)\;=\;\mathrm{sign}\left(\sigma_{x2}^{\prime }\right)\cdot \max(\left|\sigma_{x2}^{\prime }\right|-\frac{\lambda }{L},0)\;=\;\left\{\begin{array}{cc} \sigma_{x2}^{\prime }+\frac{\lambda }{L} & \sigma_{x2}^{\prime }\leq -\frac{\lambda }{L}\\ 0 & -\frac{\lambda }{L}\leq \sigma_{x2}^{\prime }\leq +\frac{\lambda }{L}\\ \sigma_{x2}^{\prime }-\frac{\lambda }{L} & \sigma_{x2}^{\prime }\geq +\frac{\lambda }{L} \end{array}\right.,$$

Non-monotonic backtracking is then employed to mitigate fluctuations in the objective function per Equation (29), thereby balancing convergence stability and global exploration. Within each iteration $\sigma_{x1}^{\prime }$, $\sigma_{x2}^{\prime }$ and $\sigma_{x3}^{\prime }$ denote successively computed scattering-coefficient vectors.

TwIST excels in TCAI applications due to its sparsity-promoting shrinkage, non-monotonic backtracking, and robustness to ill-conditioning compared with least-squares minimization (LSM), Bayesian sparse learning (SBL), and Tikhonov regularization, which are representative benchmarks for TCAI [25,30]. Unlike least-squares minimization (LSM), TwIST yields sparse solutions without matrix inversion. Compared to Bayesian sparse learning (SBL), TwIST avoids high-dimensional covariance updates, converging rapidly via matrix-vector products and element-wise operations. Relative to Tikhonov regularization, TwIST’s *L*_1_-norm shrinkage produces exact zeros for noise suppression without over-smoothing or matrix inversions.

### 4.2. Variation in Target Reconstruction with Imaging Range

The column and row correlations of RSM ***S_t_*** exhibit dependence on the imaging range *L* characterized by $Lc/f_{0}a_{C}a_{E}$, as shown in Figure 13. The spatial and temporal independences of RSM ***S_t_*** per Equations (10) and (11) are analyzed as a function of the imaging range *L* as depicted in Figure 14a, showing extrema when $Lc/f_{0}a_{C}a_{E}$ is an integer. Figure 14b demonstrates analogous range-dependent variation in RSM ***S_t_***’s effective rank. The imaging quality assessment via PSI and RIE per Equations (15) and (16) as illustrated in Figure 15 confirms a degraded performance with a low PSI and high RIE at integer $Lc/f_{0}a_{C}a_{E}$ values. In Figure 15a, the black dashed line represents the threshold of PSI = 1, with values of PSI > 1 indicating successful imaging [19].

Figure 16 demonstrates the original imaging target on the discretized imaging plane, as shown in Figure 3. The numerical simulation results are averaged over 20 trials in this research with the regularization parameter *λ* = 0.1 and the relaxation parameter *η* = 0.5. Some typical imaging results obtained by TwIST as demonstrated in Figure 17 indicate that the imaging quality does not deteriorate monotonically with an increasing imaging range. The imaging results analyzed in Section 3.1 corresponding to imaging ranges *L* = 2.5 m and *L* = 6.7 m are illustrated in Figure 17d,h, which indicates that a high degree of column correlation, as shown in Figure 4 and Figure 6, significantly degrades imaging quality. A mechanism similar to that shown in Figure 17c,f provides evidence that elevated row correlation at the imaging range *L* = 2.2333 m can also impair image quality. The values of PSI and RIE corresponding to TwIST-reconstructed images in Figure 17 are listed in Table 4.

However, in the imaging range of *L* < 1.6 m, the integer condition $Lc/f_{0}a_{C}a_{E}$ exerts limited influence on spatiotemporal independence due to dominant near-field effects that induce significant phase deviations from minute lateral distance variations. The essence of the near-field effects is the violation of the far-field approximation conditions of *L* >> |*x_a_* − *x_k_*| and *L* >> |*y_a_* − *y_k_*|. Consequently, the phase difference $\phi_{a}\left(k_{1},k_{2}\right)$ cannot be accurately evaluated using the approximate method in Equation (21), and similarly, $\phi_{k}\left(a_{1},a_{2}\right)$ cannot be assessed via Equation (26). Under these significant near-field effects, $\phi_{a}\left(k_{1},k_{2}\right)$ fails to concentrate into a narrow range, and $\phi_{k}\left(a_{1},a_{2}\right)$ can hardly cluster around discrete values, which effectively mitigates the degradation of imaging quality that would otherwise occur at integer $Lc/f_{0}a_{C}a_{E}$ in the far field. Exemplified at the imaging range *L* = 1.34 m with $Lc/f_{0}a_{C}a_{E}\;=\;6$ an integer, the two most strongly correlated columns of column 573 in RSM ***S_t_*** are columns 567 and 843, with the column correlation coefficients 0.5687 and 0.5746, respectively. And the row and column index differences between their corresponding scatterers on the imaging plane are Δ*k_M_*(573, 567) = 0, Δ*k_N_*(573, 567) = −6 and Δ*k_M_*(573, 843) = 6, Δ*k_N_*(573, 843) = 0, respectively. The phase difference vectors ***ϕ_a_***(*k*_1_ = 573, *k*_2_) between their corresponding scatterers on the imaging plane as shown in Figure 18 lack the concentration observed in Figure 5. Analogously, for the row correlation, the relative difference per Equation (25) is $\Delta \phi_{k}\left(a_{1},a_{2}\right)\;=\;1.0472\;\mathrm{rad}\;=\;2\pi /6$, but the path length-induced phase difference vector $\phi_{k}\left(a_{1},a_{2}\right)$ per Equation (24) spans nearly the entire [0, 2π] range for the scatterers with indices exceeding 1575, as evidenced in Figure 19, contrasting sharply with the clustered distribution at the imaging range *L* = 2.2333 m, as shown in Figure 10a. Consequently, insufficient column and row correlations as shown in Figure 13a at the imaging range *L* = 1.34 m preserve imaging quality, as shown in Figure 17a. However, the marginal enhancement of the spatiotemporal independence emerges as shown in Figure 20, since the attenuation of the near-field effects allows the concentration of $\phi_{a}\left(k_{1},k_{2}\right)$ per Equation (19) and the clustering of $\phi_{k}\left(a_{1},a_{2}\right)$ per Equation (24) when the effect of the integer $Lc/f_{0}a_{C}a_{E}$ intensifies gradually. When the imaging range *L* = 1.5633 m with $Lc/f_{0}a_{C}a_{E}\;=\;7$, the impact of the spatiotemporal independence deterioration on the target reconstruction imaging becomes discernible, as shown in Figure 17b.

These findings highlight an important practical implication: imaging within the near-field to far-field transition zone offers a stable and high-quality operational regime. By avoiding the oscillatory performance degradation typically induced by specific integer values of $Lc/f_{0}a_{C}a_{E}$ in the far field, this approach enhances the robustness of the system design and ensures consistent reconstruction fidelity across a varying imaging range. Beyond *L* > 1.6 m, the effect of whether $Lc/f_{0}a_{C}a_{E}$ is an integer becomes a dominant factor influencing the imaging quality.

When the imaging range reaches *L* = 2.68 m with $Lc/f_{0}a_{C}a_{E}\;=\;12$ an integer, the spatiotemporal independence of RSM ***S_t_*** peaks across *L* < 20 m, while the effective rank minimizes. For example, column 1365 of RSM ***S_t_*** is strongly correlated with 15 columns with the correlation coefficients above 0.7, as shown in Figure 21. Their corresponding scatterers reside on the imaging plane with the row and column index differences Δ*k_M_* and Δ*k_N_* of −12, 0, 12 or 24 determined by ${\left.Lc/f_{0}a_{C}a_{E}\right|}_{L\;=\;2.68\;m}$, which is consistent with the theoretical predictions in Table 5. For row correlation, the precise integer ${\left.Lc/f_{0}a_{C}a_{E}\right|}_{L\;=\;2.68\;m}$ causes the path length-induced phase vector $\phi_{k}\left(a_{1},a_{2}\right)$ per Equation (24) to cluster near discrete values spaced by $\Delta \phi_{k}\left(a_{1},a_{2}\right)\;=\;0.5236\;\mathrm{rad}\;=\;2\pi /12$ per Equation (25), as shown in Figure 22. As shown in Figure 13b, the column correlation comprises multiple extremal sets, and the level of the row correlation differs from that at *L* = 1.34 m. However, the number of extremal sets in the correlation column decreases as *L* increases.

The criterion $Lc/f_{0}a_{C}a_{E}$ increases monotonically with the imaging range *L*. At *L* = 5.025 m, $Lc/f_{0}a_{C}a_{E}\;=\;22.5\;=\;0.5M_{E}\;=\;0.5N_{E}$. For *L* < 5.025 m with $Lc/f_{0}a_{C}a_{E}<0.5M_{E}\;=\;0.5N_{E}$, columns in RSM ***S_t_*** corresponding to any two scatterers with the row and column index differences of $m\prime Lc/f_{0}a_{C}a_{E}$ (*m*′ = 0, 1, 2, …) on the imaging plane exhibit strong correlations. Consequently, column correlation distributions contain at least three extremal value sets with each set corresponding to a distinct *m′*, as shown in Figure 13b, degrading the imaging performance characterized by sustained low PSI and high RIE even under successful imaging conditions, as shown in Figure 15. For example, at *L* = 4.1317 m with $Lc/f_{0}a_{C}a_{E}\;=\;18.5$ and $2Lc/f_{0}a_{C}a_{E}\;=\;37$, column correlation extremes remain near unity, as shown in Figure 23, which severely degrades image quality, as shown in Figure 17e. As the imaging range approaches *L* = 5.025 m, PSI and RIE improve progressively, as shown in Figure 15. This improvement arises because in the imaging range 5.025 m < *L* < 10.05 m with $0.5M_{E}\;=\;0.5N_{E}<Lc/f_{0}a_{C}a_{E}<M_{E}\;=\;N_{E}$, only columns in RSM ***S_t_*** corresponding to the scatterer pairs with the row and column index differences of either 0 or $Lc/f_{0}a_{C}a_{E}$ as $2Lc/f_{0}a_{C}a_{E}>M_{E}\;=\;N_{E}$ on the imaging plane exhibit strong correlations, reducing the probability of strongly correlated columns occurring in the RSM matrix ***S_t_***. Column correlation distributions thus contain only two extremal sets of *m*′ = 0 and 1, as shown in Figure 6. The transition in the column correlation as the imaging range *L* crosses the threshold of 5.025 m is depicted in Figure 24. In Figure 24a there are three types of extreme values sets corresponding to *m′* = 0, 1 and 2, while in Figure 24b there are only two types of extreme values series corresponding to *m′* = 0 and 1. Thus, the number of strongly correlated columns in RSM ***S_t_*** decreases as *L* increases. And the oscillation amplitude of both spatiotemporal independence and effective rank attenuate with the imaging range for *L* > 2.68 m as shown in Figure 14.

At *L* = 5.025 m with $Lc/f_{0}a_{C}a_{E}\;=\;22.5$, any two columns of RSM ***S_t_*** corresponding to scatterer pairs whose column and row index differences Δ*k_M_* and Δ*k_N_* on the imaging plane are both around 22.5, i.e., either 22 or 23, are considerably correlated. For example, column 1569 correlates strongly with eight columns, as shown in Table 6, in which Δ*k_M_* and Δ*k_N_* = −23, −22, or 0. However, the non-integer ${\left.Lc/f_{0}a_{C}a_{E}\right|}_{L\;=\;5.025\;m}$ suppresses column correlation extremes to below 0.41, as shown in Figure 13c. For row correlation, the path length-induced phase vector $\phi_{k}\left(a_{1},a_{2}\right)$ per Equation (24) is intrinsically periodic over the lattice array with a natural period of *M_E_* rows and *N_E_* columns of the imaging plane. Meanwhile, the value of $\phi_{k}\left(a_{1},a_{2}\right)$ also varies periodically with a period of $2\pi /\Delta \phi_{k}\left(a_{1},a_{2}\right)$. Given that the relative difference per Equation (25) is ${\left.\Delta \phi_{k}\left(a_{1},a_{2}\right)\right|}_{L\;=\;5.025\;m}\;=\;0.2793\;\mathrm{rad}$, $M_{E}\;=\;N_{E}\;=\;\left[2\pi /{\left.\Delta \phi_{k}\left(a_{1},a_{2}\right)\right|}_{L\;=\;5.025\;m}\right]\times 2$, such harmonized matching between *M_E_* = *N_E_* and $2\pi /{\left.\Delta \phi_{k}\left(a_{1},a_{2}\right)\right|}_{L\;=\;5.025\;m}$ enables $\phi_{k}\left(a_{1},a_{2}\right)$ to uniformly and exactly cover the full [0, 2π] range twice per row/column, as shown in Figure 25. This diverse distribution of $\phi_{k}\left(a_{1},a_{2}\right)$ substantially enhances the efficacy of the random discrete phase-encoding, which is evidenced by the low level of the row correlation as shown in Figure 13c. Consequently, reduced spatiotemporal independence *γ*_space_ and *γ*_time_ and effective rank occur at *L* = 5.025 m, as shown in Figure 14, producing a high PSI and low RIE in Figure 15 and indicating superior TwIST reconstruction fidelity, as shown in Figure 17g.

At the critical imaging range *L* = 8.71 m, a key transition occurs. ${\left.Lc/f_{0}a_{C}a_{E}\right|}_{L\;=\;8.71\;m}\;=\;39$ implies that columns in RSM ***S_t_*** corresponding to scatterer pairs with the row and column index differences 0 or 39 exhibit near-unity correlation, as shown in Figure 26. However, the 16 dominant scatterers with *σ′* = 1 are constrained to the column and row index range of [7, 39] on the imaging plane, as shown in Figure 3. Given ${\left.7+Lc/f_{0}a_{C}a_{E}\right|}_{L\;=\;8.71\;m}\;=\;46>M_{E}\;=\;N_{E}$ and ${\left.39-Lc/f_{0}a_{C}a_{E}\right|}_{L\;=\;8.71\;m}\;=\;0$, columns corresponding to scatterers with *σ′* = 1 are never considerably correlated with others. Strong correlations exist only among dummy scatterers with *σ*′ = 0. This potentially enables dominant scatterer discrimination without interference from other scatterers in the target reconstruction process. Consequently, imaging results obtained via LSM, SBL, and Tikhonov regularization significantly degrade as shown in Figure 27a–c due to the presence of strongly correlated columns in RSM ***S_t_***, while TwIST maintains superior quality as shown in Figure 17i. Thus, the strong correlation between columns of RSM ***S_t_*** corresponding to dummy scatterers with *σ*′ = 0 never impairs TwIST’s performance, whereas that corresponding to dominant scatterers with *σ*′ = 1 compromises it. The abrupt disappearance of strong correlation between the RSM’s columns corresponding to scatterers with *σ′* = 1 renders the imaging quality at *L* = 8.71 m an extremum in Figure 15. For *L* > 8.71 m, the columns of RSM ***S_t_*** corresponding to the dominant scatterers with *σ*′ = 1 on the imaging plane are no longer strongly correlated, yielding PSI >1 and consistently low RIE for TwIST, as shown in Figure 15. As a result, for this *L* > 8.71 m, TwIST outperforms LSM, SBL, and Tikhonov regularization as shown in Figure 28.

As imaging range *L* increases, the phase difference vector ***ϕ_a_***(*k*, *k* + 1) per Equation (19) between wavefronts from each coding element of the coded aperture to the directly adjacent scatterers on the imaging plane decreases, where 1 < *k* < *K* − 1. This results in enhanced correlation between their corresponding columns in RSM ***S_t_***. At $L\;=\;N_{E}f_{0}a_{C}a_{E}/c\;=\;10.05\;m$ with $Lc/f_{0}a_{C}a_{E}\;=\;45$, the imaging plane comprising *M_E_* × *N_E_* = 45 × 45 scatterers contains no scatterers with the column and row index differences of ${\left.Lc/f_{0}a_{C}a_{E}\right|}_{L\;=\;10.05\;m}$. Consequently, the column correlation remains below 0.37 as Figure 13d, in which column 1012, for example, correlates significantly with eight columns as demonstrated in Figure 29. Their corresponding scatterers on the imaging plane are all directly adjacent to those corresponding to column 1012, as illustrated in Table 7. The row correlation analysis at *L* = 10.05 m is similar to that at *L* = 5.025 m. The relative difference per Equation (25) is ${\left.\Delta \phi_{k}\left(a_{1},a_{2}\right)\right|}_{L\;=\;10.05\;m}\;=\;0.1396\;\mathrm{rad}$ with $M_{E}\;=\;N_{E}\;=\;2\pi /{\left.\Delta \phi_{k}\left(a_{1},a_{2}\right)\right|}_{L\;=\;10.05\;m}$. For the path length-induced phase difference vector $\phi_{k}\left(a_{1},a_{2}\right)$ per Equation (24), this harmonized matching enables the uniform and exact coverage of the full [0, 2π] range as shown in Figure 30, which results in low row correlation, as shown in Figure 13d. Thus, TwIST achieves excellent reconstruction quality, as shown in Figure 17j.

For imaging ranges *L* > 10.05 m, the mutual correlation between RSM′s columns corresponding to adjacent scatterer pairs progressively increases. Concurrently, the relative difference $\Delta \phi_{k}\left(a_{1},a_{2}\right)$ per Equation (25) becomes too small, such that $2\pi /\Delta \phi_{k}\left(a_{1},a_{2}\right)>M_{E}\;=\;N_{E}$. This mismatch prevents the path length-induced phase difference vector $\phi_{k}\left(a_{1},a_{2}\right)$ per Equation (24) from spanning the full [0, 2π] range, creating a pronounced phase deficit. At *L* = 17 m, the column correlation extrema rise close to 0.7 as shown in Figure 13e, which is significantly higher than that at *L* = 10.05 m. The distribution of ${\left.\Delta \phi_{k}\left(a_{1},a_{2}\right)\right|}_{L\;\;=\;\;17\;m}$ per Equation (25) exhibits a distinct void as shown in Figure 31. This void serves as a fixed phase-encoding imparted from the specific coding element of the coded aperture to the particular scatterer on the imaging plane, thereby diminishing the efficacy of the random discrete phase-encoding. Consequently, the row correlation of RSM ***S_t_*** at *L* = 17 m as shown in Figure 13e is much greater than that at *L* = 10.05 m, as shown in Figure 13d. TwIST reconstruction at *L* = 17 m as shown in Figure 17k demonstrates the 1-cm feature resolvability threshold, while the imaging result at *L* = 18 m yields insufficient resolution, as shown in Figure 17l.

## 5. Conclusions

This study employs TwIST to analyze the effect of the imaging range on TCAI performance, revealing that TCAI performance varies non-monotonically with the imaging range *L*. The spatiotemporal independence of RSM ***S_t_*** is fundamentally governed by the phase relationships from the coding elements of the coded aperture to the scatterers on the imaging plane determined by the criterion $Lc/f_{0}a_{C}a_{E}$, exhibiting strong coupling between the column and row correlations. Columns in RSM ***S_t_*** corresponding to scatterer pairs with row and column index differences near integer multiples (including 0) of $Lc/f_{0}a_{C}a_{E}$ show non-negligible mutual correlation, peaking when $Lc/f_{0}a_{C}a_{E}$ is a precise integer. For row correlation, in the case with $Lc/f_{0}a_{C}a_{E}$ an integer, the path length-induced phase difference vector $\phi_{k}\left(a_{1},a_{2}\right)$ from all adjacent coding element pairs of the coded aperture to each scatterer clusters at sparsely spaced discrete values, thus undermining random phase-encoding efficacy and impairing temporal independence. Therefore, the image quality deteriorates notably when $Lc/f_{0}a_{C}a_{E}$ is a precise integer. However, a slight change in *L*, sufficient to make $Lc/f_{0}a_{C}a_{E}$ a non-integer, immediately restores high-fidelity imaging.

As *L* increases, two degradation mechanisms emerge: diminished phase differences between directly adjacent scatterers amplify column correlations, while developing phase voids of $\phi_{k}\left(a_{1},a_{2}\right)$ further compromise random discrete phase-encoding efficacy. These jointly drive a monotonic deterioration in spatiotemporal independence and a corresponding monotonic degradation of imaging quality for the imaging range $L>N_{E}f_{0}a_{C}a_{E}/c$. Crucially, strong correlations between columns corresponding to dominant scatterers with *σ*′ = 1, rather than dummy scatterers with *σ*′ = 0, critically impair reconstruction. TwIST outperforms LSM, SBL and Tikhonov regularization precisely when strong correlations occur among RSM’s columns exclusively corresponding to dummy scatterers.

In theory, the azimuth resolution of SAR is independent of the imaging range, achieved through the synthetic aperture formed by platform motion. In stark contrast, azimuth resolution in TCAI exhibits a strong and non-monotonic dependence on the imaging range *L*. This behavior is governed by the phase relationship determined by $Lc/f_{0}a_{C}a_{E}$. This phenomenon, arising from the wavefront coding principles inherent to a static aperture, underscores a unique physical mechanism in TCAI that has no counterpart in conventional SAR imaging. These findings highlight that reducing column and row correlations in RSM *S_t_* is essential for enhancing TCAI performance. Practical approaches include increasing signal bandwidth (e.g., linear frequency modulation and stepped-frequency continuous wave techniques widely used in conventional radar technology) and optimized coding strategies promoted by dynamically programmable metasurfaces.

Building upon the findings of this study, several promising directions emerge for future research on TCAI. First, employing deep learning and AI technologies, such as convolutional neural networks (CNNs) and generative adversarial networks (GANs), could significantly enhance image reconstruction quality and efficiency. Optimized coding strategies and adaptive algorithms could be integrated to mitigate performance degradation at critical ranges. Second, further advances in sparse reconstruction and compressed sensing could improve performance in complex scenarios, including multi-scattering and absorptive media. Moreover, it is imperative to perform experimental validation to corroborate these findings in the next phase of research.

## Figures and Tables

**Figure 1 sensors-25-05667-f001:**
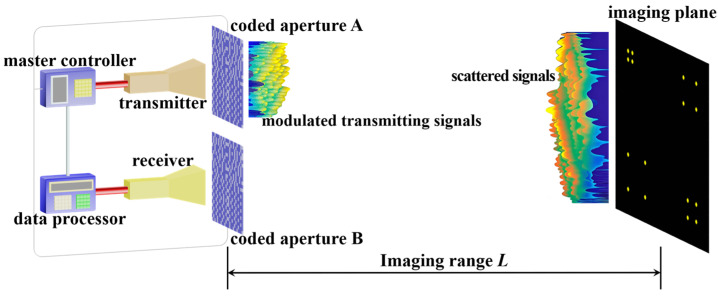
Schematic diagram of the TCAI system working in forward-looking mode.

**Figure 2 sensors-25-05667-f002:**
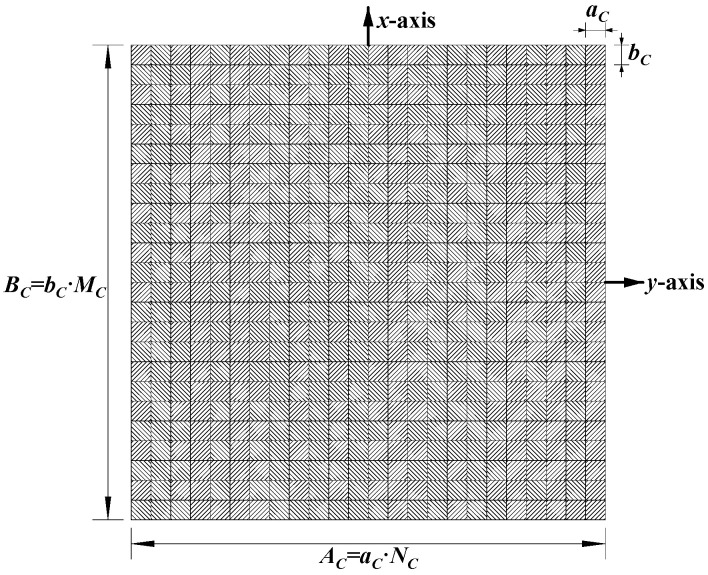
*M_C_* × *N_C_* element configuration of coded aperture.

**Figure 3 sensors-25-05667-f003:**
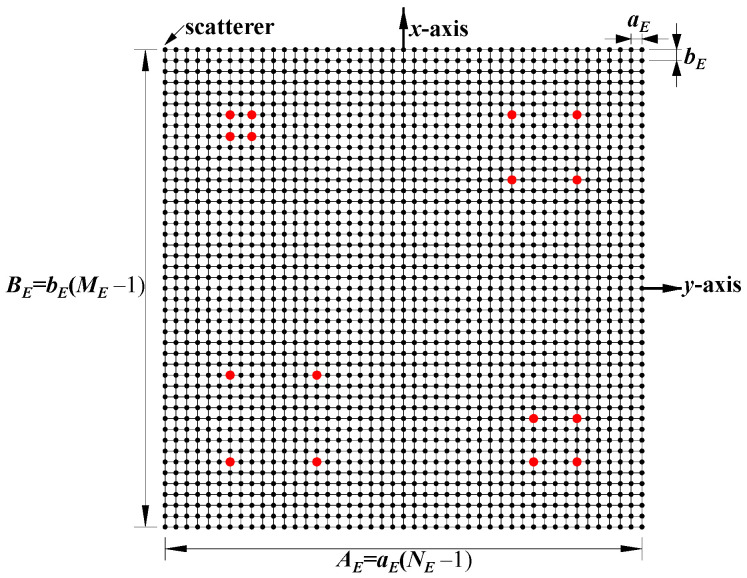
Configuration of imaging plane consisting of (*M_E_* − 1) × (*N_E_* − 1) rectangular cells with *M_E_
*× *N_E_* scatterers located exclusively at vertices. Red points represent dominant scatterers (*σ*′ = 1).

**Figure 4 sensors-25-05667-f004:**
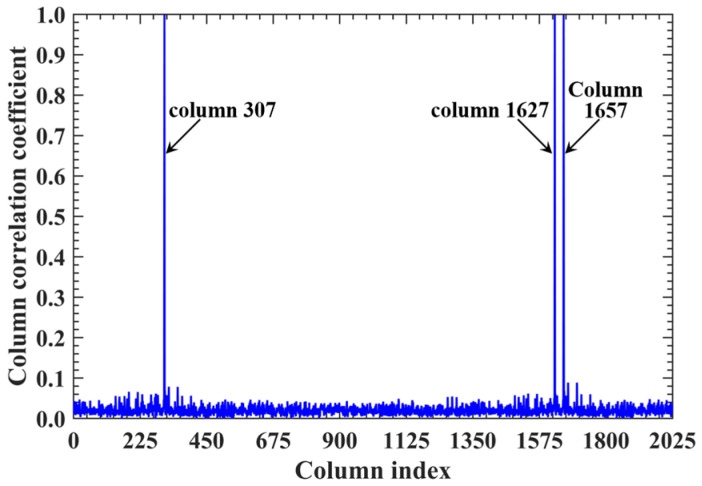
Correlation of column 277 in RSM at *L* = 6.7 m. Near-unity correlation of column 277 with columns 307, 1627, and 1657 (Table 2) stems from $Lc/f_{0}a_{C}a_{E}\;=\;30$, which induces the concentration of phase difference vectors between these scatterer pairs (Figure 5).

**Figure 5 sensors-25-05667-f005:**
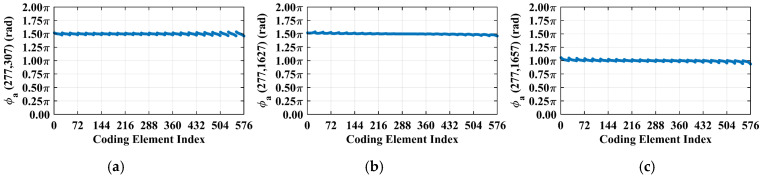
Phase difference vectors (**a**) $\phi_{a}\left(277,307\right)$, (**b**) $\phi_{a}\left(277,1627\right)$, and (**c**) $\phi_{a}\left(277,1657\right)$ per Equation (19) concentrate at fixed values (~1.5π, ~1.5π, and ~π, respectively) across all coding elements. Such concentration stems from strong column correlation at *L* = 6.7 m with $Lc/f_{0}a_{C}a_{E}\;=\;30$ (Figure 4).

**Figure 6 sensors-25-05667-f006:**
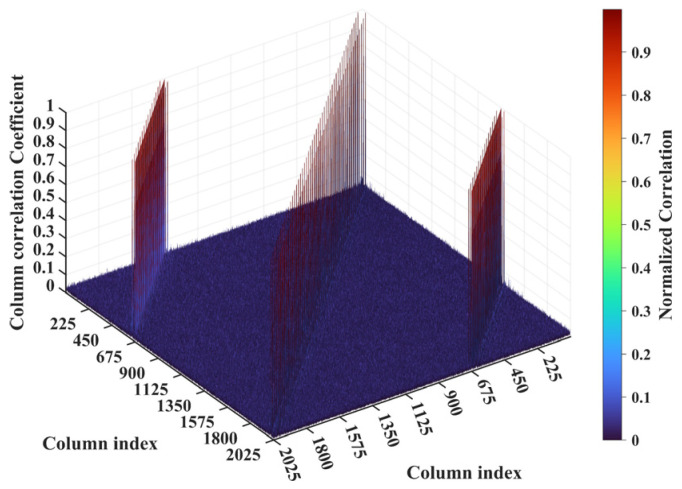
Column correlation of RSM at *L* = 6.7 m with $Lc/f_{0}a_{C}a_{E}\;=\;30$. The diagonal peak set corresponds to scatterer pairs with Δ*k_M_* = 0 and Δ*k_N_* = −30 or 30, while two off-diagonal peak sets correspond to Δ*k_M_* = −30 or 30 and Δ*k_N_* = −30, 0 or 30.

**Figure 7 sensors-25-05667-f007:**
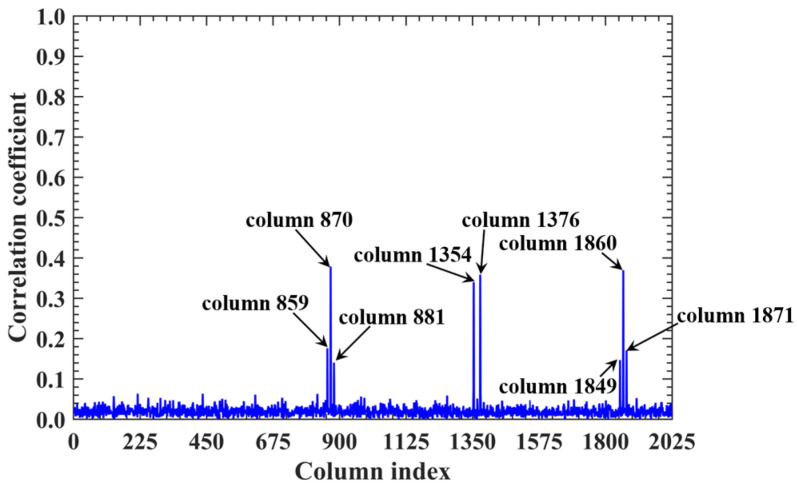
Correlation of column 1365 in RSM at *L* = 2.5 m. All peaks are suppressed below 0.4 due to non-integer ${\left.Lc/f_{0}a_{C}a_{E}\right|}_{L\;=\;2.5\;m}\;=\;11.19$, which prevents phase concentration (Figure 8). Columns exhibiting considerable correlation with column 1365 correspond to scatterers whose row and column index differences are around ${\left.Lc/f_{0}a_{C}a_{E}\right|}_{L\;=\;2.5\;m}\;=\;11.19$ (Table 3).

**Figure 8 sensors-25-05667-f008:**
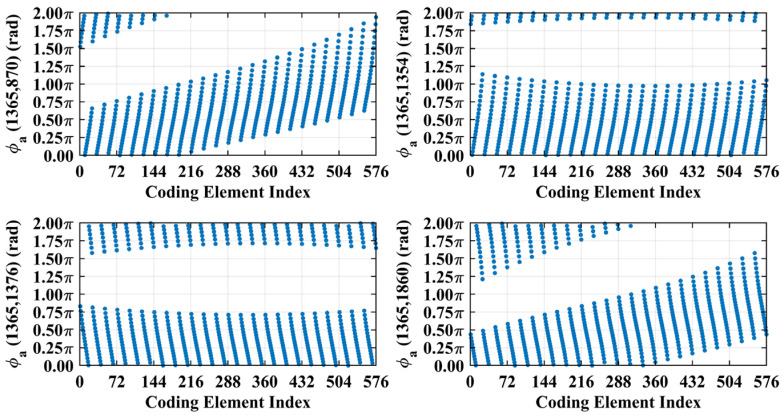
Phase difference vectors $\phi_{a}\left(1365,870\right)$, $\phi_{a}\left(1365,1354\right)$, $\phi_{a}\left(1365,1376\right)$ and $\phi_{a}\left(1365,1860\right)$ per Equation (19) for the four strongest correlated column pairs identified in Figure 7. Vector values cover a broad portion of [0, 2π] due to non-integer $Lc/f_{0}a_{C}a_{E}\;=\;11.19$ at *L* = 2.5 m.

**Figure 9 sensors-25-05667-f009:**
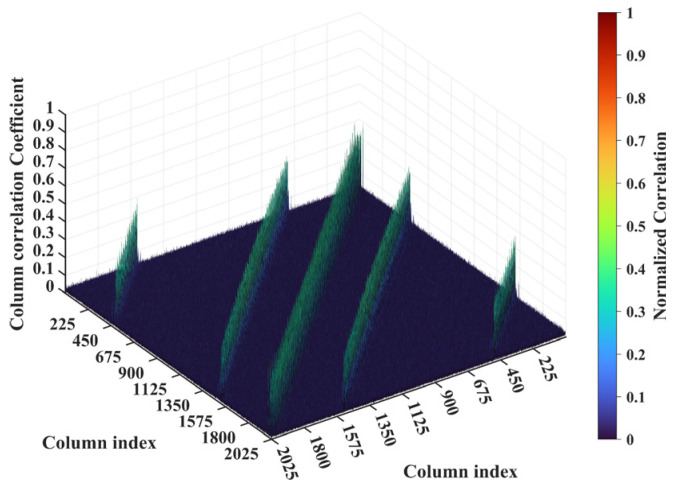
Column correlation of RSM at *L* = 2.5 m with all peaks suppressed below 0.4 due to non-integer $Lc/f_{0}a_{C}a_{E}\;=\;11.19$, which prevents the concentration of phase difference vectors $\phi_{a}\left(k_{1},k_{2}\right)$ (Figure 8). Column pairs with considerable correlation correspond to scatterers whose row and column index differences are both around $Lc/f_{0}a_{C}a_{E}\;=\;11.19$ (Table 3).

**Figure 10 sensors-25-05667-f010:**
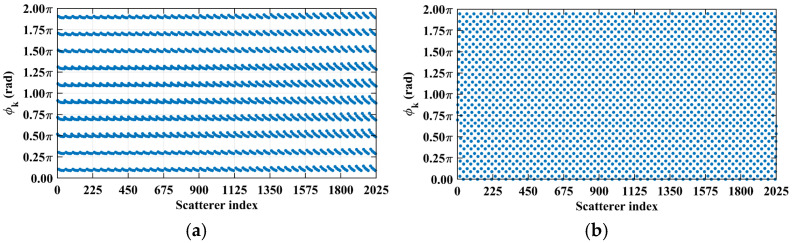
Comparison of path length-induced phase difference vectors $\phi_{k}\left(a_{1},a_{2}\right)$ per Equation (24). (**a**) At *L* = 2.2333 m with integer $Lc/f_{0}a_{C}a_{E}\;=\;10$, a value of $\phi_{k}\left(a_{1},a_{2}\right)$ clusters around discrete values spaced at $\Delta \phi_{k}\left(a_{1},a_{2}\right)\;=\;2\pi /10$. (**b**) At *L* = 5.0 m with non-integer $Lc/f_{0}a_{C}a_{E}\;=\;22.3881$, the value of $\phi_{k}\left(a_{1},a_{2}\right)$ uniformly covers nearly the entire [0, 2π].

**Figure 11 sensors-25-05667-f011:**
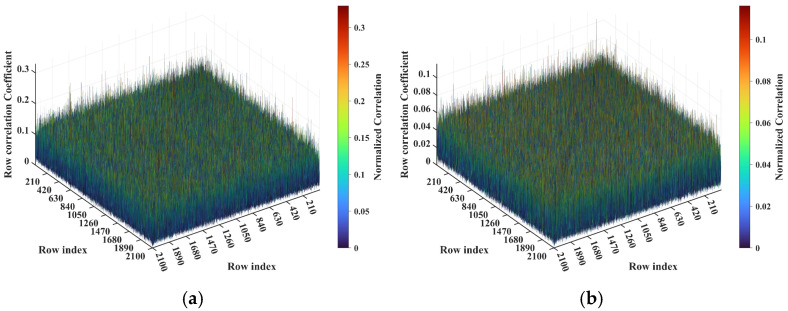
Row correlation in RSM: (**a**) At *L* = 2.2333 m with integer $Lc/f_{0}a_{C}a_{E}\;=\;10$, the maximum of the row correlation exceeds 0.2 due to clustering of the $\phi_{k}\left(a_{1},a_{2}\right)$ value (Figure 10a). (**b**) At *L* = 5.0 m with non-integer $Lc/f_{0}a_{C}a_{E}\;=\;22.3881$, the maximum of the row correlation stays below 0.1 due to uniform coverage over [0, 2π] of the $\phi_{k}\left(a_{1},a_{2}\right)$ value (Figure 10b).

**Figure 12 sensors-25-05667-f012:**
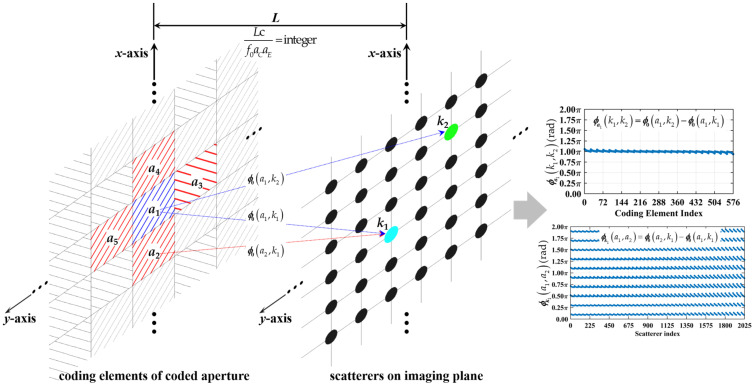
Mechanism diagram of integer $Lc/f_{0}a_{C}a_{E}$ leading to strong column and row correlation. *a*_2_-, *a*_3_-, *a*_4_- and *a*_5_-th coding elements are directly adjacent to *a*_1_-th one. *k*_1_- and *k*_2_-th scatterers with row and column index differences of integer multiples (including 0) of $Lc/f_{0}a_{C}a_{E}$.$\phi_{0}\left(a_{1},k_{1}\right)$, $\phi_{0}\left(a_{1},k_{2}\right)$ and $\phi_{0}\left(a_{2},k_{1}\right)$ are path-induced phase difference from *a*_1_ to *k*_1_, *a*_1_ to *k*_2_, and *a*_2_ to *k*_1_ per Equation (18), respectively. Attributed to $Lc/f_{0}a_{C}a_{E}$ being precisely an integer, $\phi_{a_{1}}\left(k_{1},k_{2}\right)\;=\;\phi_{0}\left(a_{1},k_{2}\right)-\phi_{0}\left(a_{1},k_{1}\right)$ per Equation (19) concentrates within a specific narrow numerical range, and $\phi_{k_{1}}\left(a_{1},a_{2}\right)\;=\;\phi_{0}\left(a_{2},k_{1}\right)-\phi_{0}\left(a_{1},k_{1}\right)$ per Equation (24) clusters around sparsely spaced, highly discrete specific values determined by imaging range. Such clustering behavior is shared by $\phi_{k_{1}}\left(a_{1},a_{3}\right)\;=\;\phi_{0}\left(a_{3},k_{1}\right)-\phi_{0}\left(a_{1},k_{1}\right)$, $\phi_{k_{1}}\left(a_{1},a_{4}\right)\;=\;\phi_{0}\left(a_{4},k_{1}\right)-\phi_{0}\left(a_{1},k_{1}\right)$, $\phi_{k_{1}}\left(a_{1},a_{5}\right)\;=\;\phi_{0}\left(a_{5},k_{1}\right)-\phi_{0}\left(a_{1},k_{1}\right)$ and $\phi_{k_{2}}\left(a_{1},a_{2}\right)\;=\;\phi_{0}\left(a_{2},k_{2}\right)-\phi_{0}\left(a_{1},k_{2}\right)$, $\phi_{k_{2}}\left(a_{1},a_{3}\right)\;=\;\phi_{0}\left(a_{3},k_{2}\right)-\phi_{0}\left(a_{1},k_{2}\right)$, $\phi_{k_{2}}\left(a_{1},a_{4}\right)\;=\;\phi_{0}\left(a_{4},k_{2}\right)-\phi_{0}\left(a_{1},k_{2}\right)$, $\phi_{k_{2}}\left(a_{1},a_{5}\right)\;=\;\phi_{0}\left(a_{5},k_{2}\right)-\phi_{0}\left(a_{1},k_{2}\right)$. 1 ≤ *a*_1_, *a*_2_, *a*_3_, *a*_4_, *a*_5_ ≤ *M_C_* × *N_C_*, and 1 ≤ *k*_1_, *k*_2_ ≤ *K* = *M_E_* × *N_E_*.

**Figure 13 sensors-25-05667-f013:**
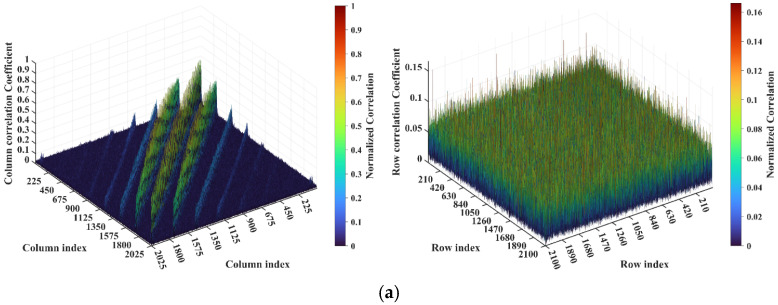
Evolution of column and row correlation of RSM with imaging range *L*: (**a**) *L* = 1.34 m with $Lc/f_{0}a_{C}a_{E}\;=\;6$; (**b**) *L* = 2.68 m with $Lc/f_{0}a_{C}a_{E}\;=\;12$; (**c**) *L* = 5.025 m with $Lc/f_{0}a_{C}a_{E}\;=\;22.5$; (**d**) *L* = 10.05 m with $Lc/f_{0}a_{C}a_{E}\;=\;45$; and (**e**) $L\;=\;17\;m>N_{E}f_{0}a_{C}a_{E}/c$. Correlations exhibit non-monotonic behavior with respect to *L*. Peak correlations occur when $Lc/f_{0}a_{C}a_{E}$ is a precise integer (e.g., *L* = 1.34 m, 2.68 m, 10.05 m). As *L* and consequently $Lc/f_{0}a_{C}a_{E}$ increase, the number of distinct extremal sets in column correlation distribution decreases.

**Figure 14 sensors-25-05667-f014:**
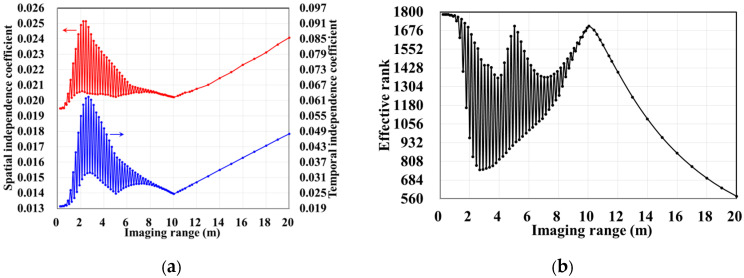
Evolution of (**a**) spatial and temporal independence functions *γ*_space_ and *γ*_time_ per Equations (10) and (11); (**b**) effective rank with imaging range *L*. Spatiotemporal independence (*γ*_space_ and *γ*_time_) and effective rank exhibit non-monotonic evolution for *L* < 10.05 m, with deterioration occurring when $Lc/f_{0}a_{C}a_{E}$ is a precise integer. For *L* > 10.05 m, spatiotemporal independence (*γ*_space_ and *γ*_time_) and effective rank monotonically degenerate due to diminishing inter-scatterer phase differences and the emergence of phase deficits in the encoding process.

**Figure 15 sensors-25-05667-f015:**
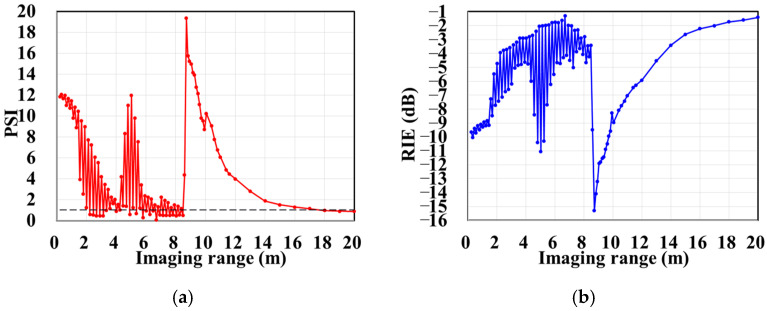
Variation in reconstruction imaging quality using TwIST with imaging range *L*: (**a**) PSI with black dashed line representing threshold PSI = 1 and PSI > 1 indicating successful imaging; (**b**) RIE with smaller value indicating superior imaging quality.

**Figure 16 sensors-25-05667-f016:**
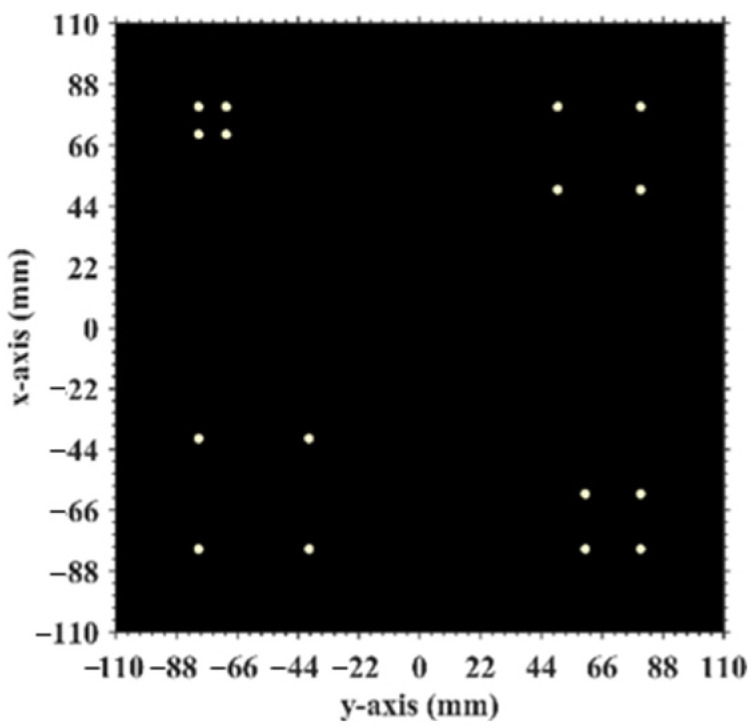
Original imaging target which is discretized into the imaging plane (Figure 3).

**Figure 17 sensors-25-05667-f017:**
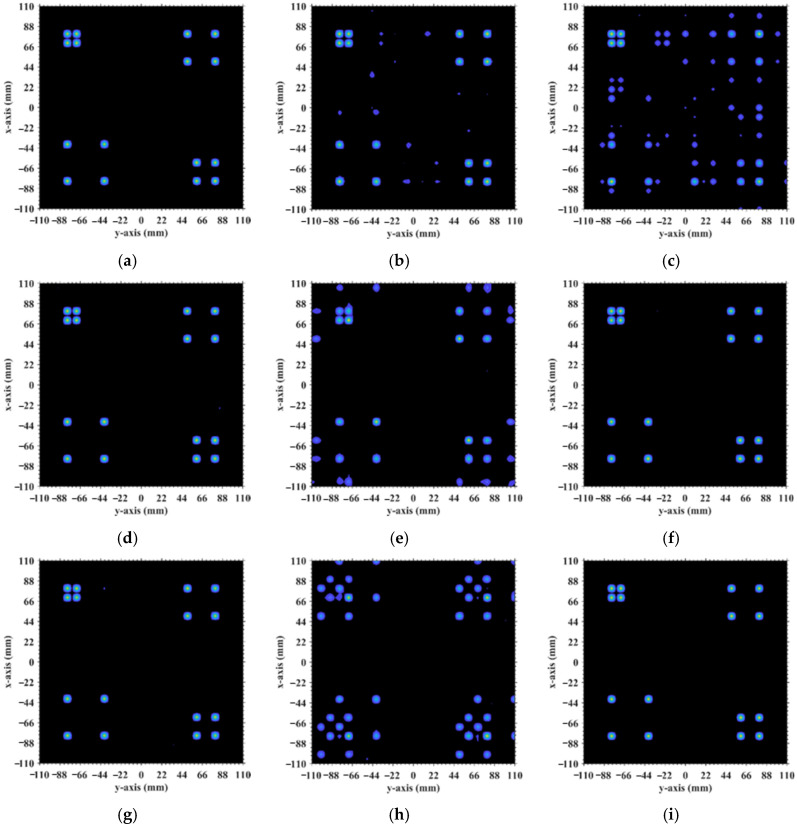
TwIST-reconstructed images: (**a**) At *L* = 1.34 m with $Lc/f_{0}a_{C}a_{E}\;=\;6$; (**b**) *L* = 1.5633 m with $Lc/f_{0}a_{C}a_{E}\;=\;7$; (**c**) *L* = 2.2333 m with $Lc/f_{0}a_{C}a_{E}\;=\;10$; (**d**) *L* = 2.5 m with $Lc/f_{0}a_{C}a_{E}\;=\;11.19$; (**e**) *L* = 4.1317 m with $Lc/f_{0}a_{C}a_{E}\;=\;18.5$ and $2Lc/f_{0}a_{C}a_{E}\;=\;37$; (**f**) *L* = 5.0 m with $Lc/f_{0}a_{C}a_{E}\;=\;22.3881$; (**g**) *L* = 5.025 m with $Lc/f_{0}a_{C}a_{E}\;=\;22.5\;=\;0.5M_{E}\;=\;0.5N_{E}$; (**h**) *L* = 6.7 m with $Lc/f_{0}a_{C}a_{E}\;=\;30$; (**i**) *L* = 8.71 m with $Lc/f_{0}a_{C}a_{E}\;=\;39$; (**j**) *L* = 10.05 m with $Lc/f_{0}a_{C}a_{E}\;=\;45$; (**k**) $L\;=\;17\;m>N_{E}f_{0}a_{C}a_{E}/c$; (**l**) $L\;=\;18\;m>N_{E}f_{0}a_{C}a_{E}/c$. For *L* < 1.6 m, near-field effects preserve phase diversity and thus imaging quality (e.g., *L* = 1.34 m). At intermediate range (1.6 m < L < 5.025 m), quality degrades severely if $mLc/f_{0}a_{C}a_{E}$ is an integer (*m* is an integer, e.g., *L* = 4.1317 m). For 5.025 m < *L* < 8.71 m, imaging degradation recurs when $Lc/f_{0}a_{C}a_{E}$ is a precise integer (e.g., *L* = 6.7 m). For 8.71 m ≤ *L* < 10.05 m, TwIST reconstruction imaging succeeds at any *L* because strong correlations involve dominant scatterers (*σ*′ = 1) which are no longer present. For $L>N_{E}f_{0}a_{C}a_{E}/c\;=\;10.05\;m$, imaging quality degrades monotonically with *L* (e.g., *L* = 17 m, 18 m). High quality is achieved at non-integer $Lc/f_{0}a_{C}a_{E}$ (e.g., *L* = 2.5 m, 5.0 m, 5.025 m, 10.05 m).

**Figure 18 sensors-25-05667-f018:**
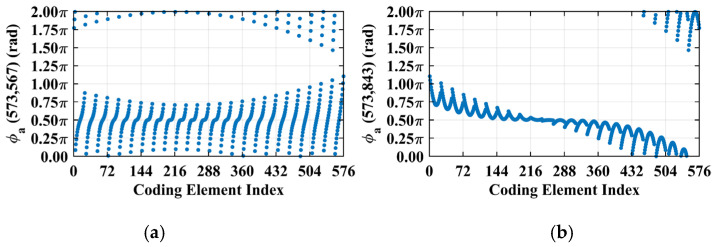
Phase difference vectors (**a**) $\phi_{a}\left(573,567\right)$ and (**b**) $\phi_{a}\left(573,843\right)$ per Equation (19) at *L* = 1.34 m with $Lc/f_{0}a_{C}a_{E}\;=\;6$. But significant near-field effects introduce large phase deviations, preventing their concentration.

**Figure 19 sensors-25-05667-f019:**
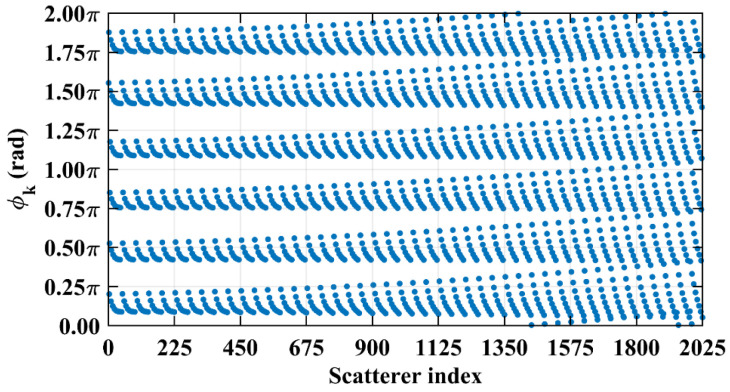
Path length-induced phase difference vector $\phi_{k}\left(a_{1},a_{2}\right)$ per Equation (24) at *L* = 1.34 m with $Lc/f_{0}a_{C}a_{E}\;=\;6$ spans nearly the entirety of [0, 2π] for scatterers with indices exceeding 1575 due to significant near-field effects.

**Figure 20 sensors-25-05667-f020:**
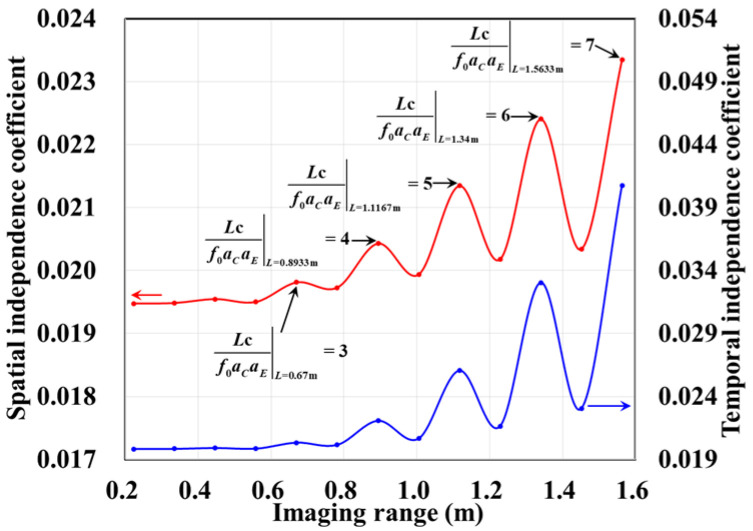
Zoom-in view of imaging range *L* < 1.6 m from Figure 14. As *L* increases, attenuation of near-field effects allows $\phi_{a}\left(k_{1},k_{2}\right)$ to concentrate and $\phi_{k}\left(a_{1},a_{2}\right)$ to cluster more severely when $Lc/f_{0}a_{C}a_{E}$ is a precise integer. This leads to increasingly pronounced extremal points in spatiotemporal independence at integer values of $Lc/f_{0}a_{C}a_{E}$.

**Figure 21 sensors-25-05667-f021:**
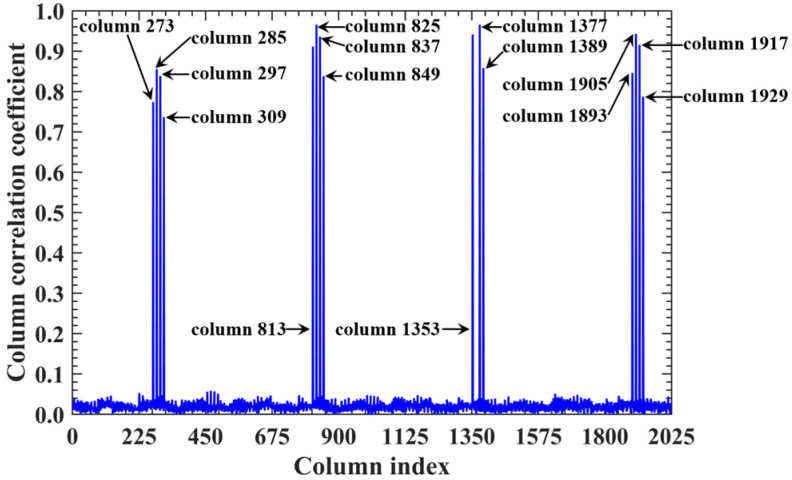
Correlation of column 1365 in RSM at *L* = 2.68 m. Fifteen strongly correlated columns with correlation coefficients above 0.7 correspond to scatterers with row and column index differences Δ*k_M_* and Δ*k_N_* of −12, 0, 12, or 24 (Table 5), as determined by ${\left.Lc/f_{0}a_{C}a_{E}\right|}_{L\;=\;2.68\;m}\;=\;12$.

**Figure 22 sensors-25-05667-f022:**
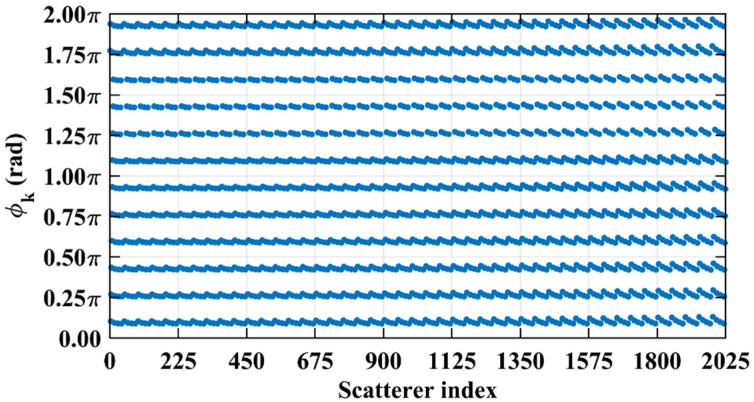
Path length-induced phase difference vector $\phi_{k}\left(a_{1},a_{2}\right)$ per Equation (24) at *L* = 2.68 m with $Lc/f_{0}a_{C}a_{E}\;=\;12$ cluster near discrete values spaced by $\Delta \phi_{k}\left(a_{1},a_{2}\right)\;=\;2\pi /12$, which is determined by the imaging range.

**Figure 23 sensors-25-05667-f023:**
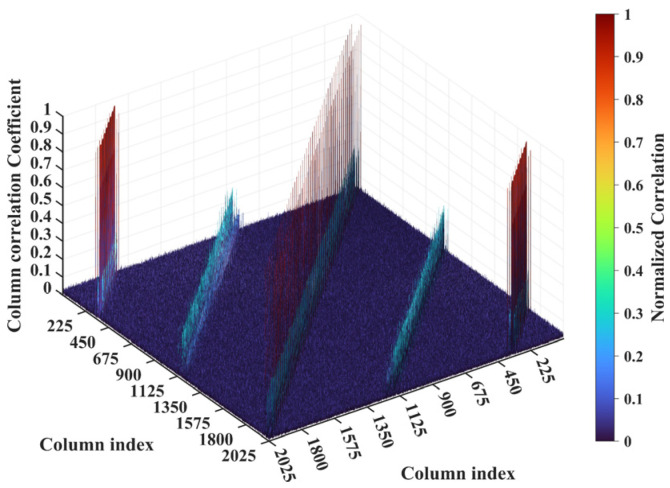
Column correlation of RSM at *L* = 2.5 m at *L* = 4.1317 m with $Lc/f_{0}a_{C}a_{E}\;=\;18.5$ and $2Lc/f_{0}a_{C}a_{E}\;=\;37$. Three peak sets of value close to unity correspond to scatterer pairs with row and column index differences for Δ*k_M_* and Δ*k_N_* of −37, 0, or 37.

**Figure 24 sensors-25-05667-f024:**
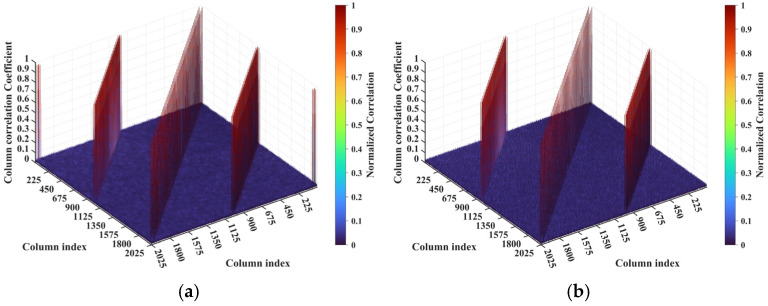
Transition of column correlation of RSM from imaging range (**a**) *L* = 4.9133 m to (**b**) *L* = 5.1367 m. For *L* = 4.9133 m with $Lc/f_{0}a_{C}a_{E}\;=\;22$, diagonal and off-diagonal peak sets correspond to scatterer pairs with row and column index differences Δ*k_M_* and Δ*k_N_* of −22, 0, or 22, while peak sets at two corners correspond to Δ*k_M_* and Δ*k_N_* of −44, 0, or 44. For *L* = 5.1367 m with $Lc/f_{0}a_{C}a_{E}\;=\;23$, the diagonal peak set corresponds to scatterer pairs with Δ*k_M_* = 0 and Δ*k_N_* = −23 or 23, while two off-diagonal peak sets correspond to Δ*k_M_* = −23 or 23 and Δ*k_N_* = −23, 0 or 23, and peak sets at corners disappear.

**Figure 25 sensors-25-05667-f025:**
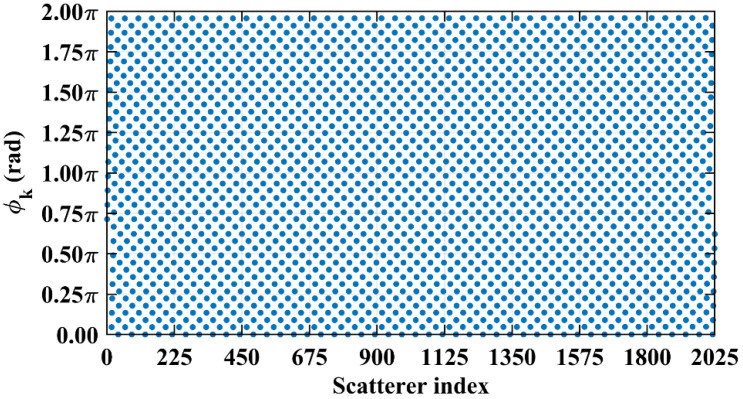
Path length-induced phase difference vector $\phi_{k}\left(a_{1},a_{2}\right)$ per Equation (24) at *L* = 5.025 m with $Lc/f_{0}a_{C}a_{E}\;=\;22.5$ uniformly and exactly covering the full [0, 2π] range.

**Figure 26 sensors-25-05667-f026:**
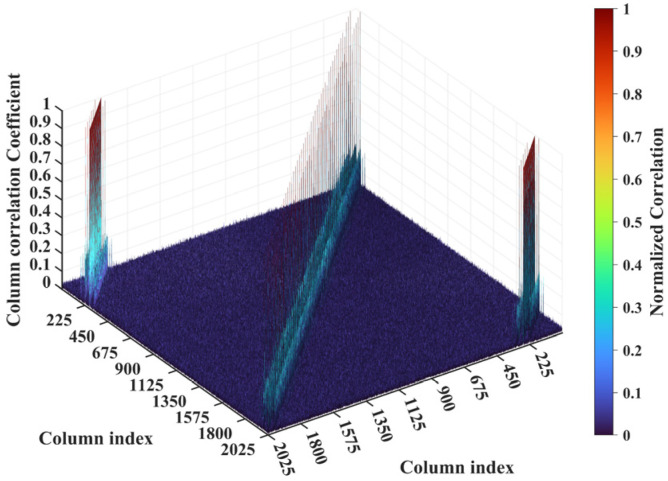
Column correlation of RSM ***S_t_*** at imaging range *L* = 8.71 m. All peak sets of value close to unity are caused by strong column correlations involving dominant scatterers (*σ*′ = 1).

**Figure 27 sensors-25-05667-f027:**
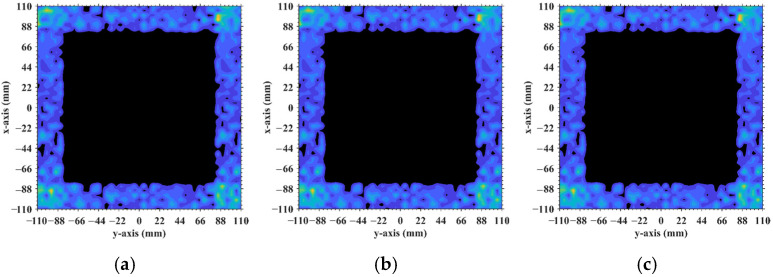
Imaging results obtained by 3 reconstruction algorithms at an imaging range of *L* = 8.71 m: (**a**) LSM; (**b**) SBL; (**c**) Tikhonov regularization, indicating that strong column correlations, even those involving dummy scatterers (*σ*′ = 0), can adversely affect imaging quality. Brighter colors represent larger magnitudes, while darker areas represent values closer to zero.

**Figure 28 sensors-25-05667-f028:**
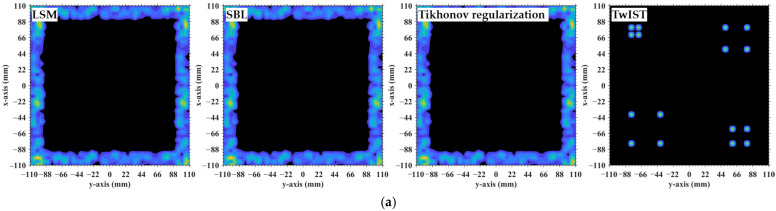
Comparison of imaging results obtained by LSM, SBL, Tikhonov regularization and TwIST: (**a**) At *L* = 8.9333 m with $Lc/f_{0}a_{C}a_{E}\;=\;40$; (**b**) at *L* = 9.8267 m with $Lc/f_{0}a_{C}a_{E}\;=\;44$; and (**c**) at $L\;=\;11.1\;m>N_{E}f_{0}a_{C}a_{E}/c$. Imaging results of LSM, SBL, and Tikhonov regularization are highly consistent with each other, yet all are inferior to those of TwIST. Brighter colors represent larger magnitudes, while darker areas represent values closer to zero.

**Figure 29 sensors-25-05667-f029:**
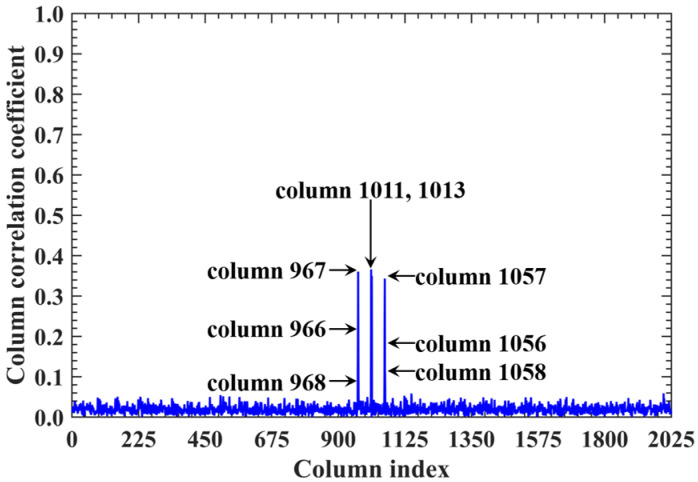
Correlation of column 1012 in RSM at *L* = 10.05 m. All columns significantly correlated with column 1012 correspond to scatterers directly adjacent to those corresponding to column 1012 (Table 7). Thus, columns significantly correlated with column 1012 are in close proximity and indistinguishable.

**Figure 30 sensors-25-05667-f030:**
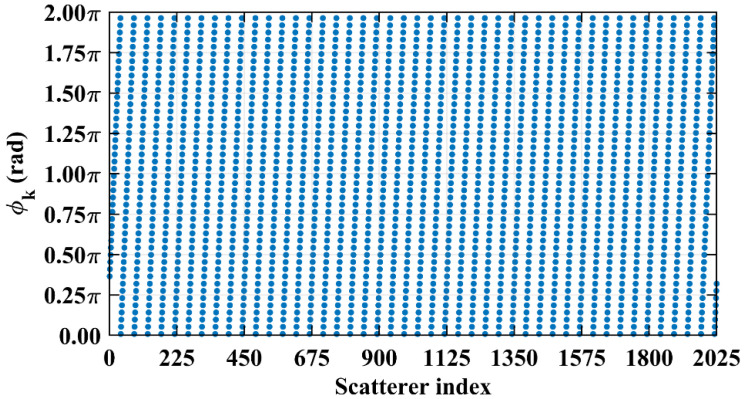
Path length-induced phase difference vector $\phi_{k}\left(a_{1},a_{2}\right)$ per Equation (24) at *L* = 10.05 m uniformly and exactly cover the full [0, 2π] range.

**Figure 31 sensors-25-05667-f031:**
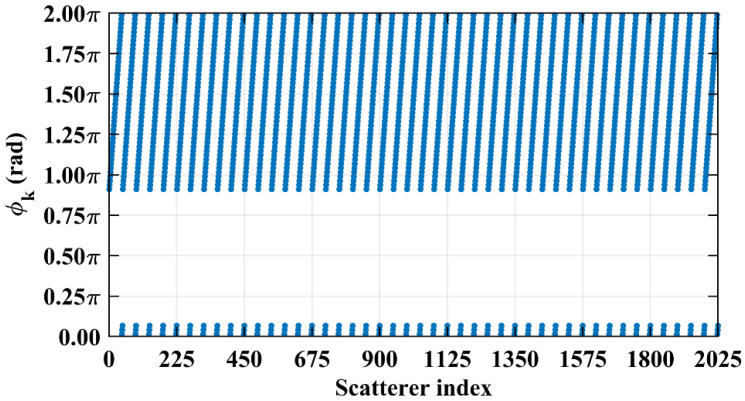
Path length-induced phase difference vector $\phi_{k}\left(a_{1},a_{2}\right)$ per Equation (24) at *L* = 17 m failed to span [0, 2π] due to a too small $\Delta \phi_{k}\left(a_{1},a_{2}\right)$ per Equation (25), creating a pronounced phase deficit.

**Table 1 sensors-25-05667-t001:** Primary parameters used in numerical simulations.

Parameter	Value
Frequency of terahertz signal (*f*_0_)	670 GHz
Sampling intervals (*T_s_*)	2 μs
Sampling number	2100
Axial inter-element pitch on coded aperture A and B $(a_{C}\times b_{C}$ each)	2 cm × 2 cm
Coding element counts on both coded apertures A and B (*M_C_* × *N_C_* each)	24 × 24
Axial spacing between adjacent scatterers on imaging plane $(a_{E}\times b_{E}$)	5 mm × 5 mm
Imaging plane size (*A_E_* × *B_E_*)	22 cm × 22 cm
SNR	20 dB

**Table 2 sensors-25-05667-t002:** Corresponding scatterers of column 277 and its strongly correlated column of RSM ***S_t_*** at the imaging range *L* = 6.7 m.

Column of RSM	Column Correlation	Corresponding Scatterer on Imaging Plane			
		Row	Column	Coordinates (mm)	Scattering Coefficient
277		7	7	(80, −80)	1
307	0.9988	7	37	(80, 70)	0
1627	0.9989	37	7	(−70, −80)	0
1657	0.9980	37	37	(−70, 70)	0

**Table 3 sensors-25-05667-t003:** Corresponding scatterers of column 1365 and its correlated column of RSM ***S_t_*** at imaging range *L* = 2.5 m.

Column of RSM	Column Correlation	Corresponding Scatterer on Imaging Plane			
		Row	Column	Coordinates (mm)	Scattering Coefficient
1365		31	15	(−40, −40)	1
859	0.1763	20	4	(15, −95)	0
870	0.3784	20	15	(15, −40)	0
881	0.1407	20	26	(15, 15)	0
1354	0.3398	31	4	(−40, −95)	0
1376	0.3586	31	26	(−40, 15)	0
1849	0.1469	42	4	(−95, −95)	0
1860	0.3695	42	15	(−95, −40)	0
1871	0.1712	42	26	(−95, 15)	0

**Table 4 sensors-25-05667-t004:** PSI and RIE corresponding to TwIST-reconstructed images in Figure 17.

Imaging Range *L* (m)	1.3400	1.5633	2.2333	2.5000	4.1317	5.0000	5.0250	6.7000	8.7100	10.0500	17.0000	18.0000
$Lc/f_{0}a_{C}a_{E}$	6	7	10	11.19	18.5	22.3881	22.5	30	39	45	$>N_{E}f_{0}a_{C}a_{E}/c$	$>N_{E}f_{0}a_{C}a_{E}/c$
PSI	8.8996	3.9215	0.5853	4.7548	1.5152	8.7623	11.9813	0.0648	19.3324	10.2469	1.1357	0.96775
RIE (dB)	−8.8408	−1.5071	−3.9503	−5.1457	−4.7530	−8.5829	−11.0658	−1.3091	−15.2905	−8.9734	−2.0035	−1.7282

**Table 5 sensors-25-05667-t005:** Corresponding scatterers of column 1365 and its correlated column of RSM ***S_t_*** at imaging range *L* = 2.68 m.

Column of RSM	Column Correlation	Corresponding Scatterer on Imaging Plane			
		Row	Column	Coordinates (mm)	Scattering Coefficient
1365		31	15	(−40, −40)	1
273	0.7738	7	3	(80, −100)	0
285	0.8553	7	15	(80, −40)	0
297	0.8386	7	27	(80, 20)	0
309	0.7362	7	39	(80, 80)	0
813	0.9110	19	3	(20, −100)	0
825	0.9656	19	15	(20, −40)	0
837	0.9362	19	27	(20, 20)	0
849	0.8379	19	39	(20, 80)	0
1353	0.9411	31	3	(−40, −100)	0
1377	0.9658	31	27	(−40, 20)	0
1389	0.8585	31	39	(−40, 80)	0
1893	0.8460	43	3	(−100, −100)	0
1905	0.9426	43	15	(−100, −40)	0
1917	0.9155	43	27	(−100, 20)	0
1929	0.7871	43	39	(−100, 80)	0

**Table 6 sensors-25-05667-t006:** Corresponding scatterers of column 1569 and its correlated column of RSM ***S_t_*** at an imaging range *L* = 5.025 m.

Column of RSM	Column Correlation	Corresponding Scatterer on Imaging Plane			
		Row	Column	Coordinates (mm)	Scattering Coefficient
1569		35	39	(−60, 80)	1
511	0.1619	12	16	(55, −35)	0
512	0.0948	12	17	(55, −30)	0
534	0.3679	12	39	(55, 80)	0
556	0.1496	13	16	(50, −35)	0
557	0.1164	13	17	(50, −30)	0
579	0.3507	13	39	(50, 80)	0
1546	0.4054	35	16	(−60, −35)	0
1547	0.3138	35	17	(−60, −30)	0

**Table 7 sensors-25-05667-t007:** Corresponding scatterers of column 1012 and its correlated column of RSM ***S_t_*** at imaging range *L* = 10.05 m.

Column of RSM	Column Correlation	Corresponding Scatterer on Imaging Plane			
		Row	Column	Coordinates (mm)	Scattering Coefficient
1012		23	22	(0, −5)	0
966	0.1505	22	21	(5, −10)	0
967	0.3606	22	22	(5, −5)	0
968	0.1048	22	23	(5, 0)	0
1011	0.3662	23	21	(0, −10)	0
1013	0.3498	23	23	(0, 0)	0
1056	0.1364	24	21	(−5, −10)	0
1057	0.3433	24	22	(−5, −5)	0
1058	0.1115	24	23	(−5, 0)	0

## Data Availability

Research data in this study is contained within the article. Further inquiries can be directed to the corresponding author.

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
