# Peer review of "Effect of Imaging Range on Performance of Terahertz Coded-Aperture Imaging"

_sensors, 2025, doi:10.3390/s25185667_

Round 1
Reviewer 1 Report
Comments and Suggestions for Authors
This paper investigates the effect of imaging range on the performance of terahertz coded-aperture imaging (TCAI). This paper discussed non-monotonic relationship between imaging quality and range with theoretical derivations and numerical simulations. The topic is clear and of innovation, which is valuable for TCAI system design and applications. The paper could be accepted after minor revision. Comments are below.
1. Table 1 provides the main simulation parameters, but the rationale for their selection is not explained. Author should briefly justify the choice of 670 GHz and SNR = 20 dB, and clarify whether these values are representative of practical scenarios.
2. The manuscript states that TwIST is better than LSM, SBL, and Tikhonov regularization, but the supporting experimental results are limited. Authors should add a comparative figure or table that quantitatively to show the differences of these algorithms.
3. In Section 4.2, the discussion of near-field effects for L < 1.6 m is a little bit brief. Authors should discuss the practical significance of the near-field–to–far-field transition region.
4. The conclusion section mainly summarizes the findings but lacks forward-looking perspectives. Authors should add a short paragraph to discuss potential future work, such as using optimized coding strategies or adaptive algorithms to mitigate performance degradation. It would be helpful for further enhancement of the application of the study.
Author Response
Manuscript Title: Effect of Imaging Range on Performance of Terahertz Coded-Aperture Imaging
Manuscript ID: sensors-3843017
Authors: Yan Teng, Haodong Yang, Xinhong Cui, Xiaoze Li, and Yanchao Shi
To: Ms. Chelsey Wang
From: ganlong1982@foxmail.com
Dear Editors and Reviewer,
Please find enclosed our revised manuscript entitled “Effect of Imaging Range on Performance of Terahertz Coded-Aperture Imaging” (Manuscript ID: sensors-3843017) for consideration in Sensors.
We would like to express our sincere gratitude to you and Reviewer for the constructive comments and valuable suggestions. These comments have been immensely helpful in improving the quality of our manuscript. We have carefully considered all the points raised and have made significant revisions to the manuscript accordingly.
Below, we provide a point-by-point response to the reviewers’ comments. All changes in the revised manuscript have been highlighted.
Comment 1: Table 1 provides the main simulation parameters, but the rationale for their selection is not explained. Author should briefly justify the choice of 670 GHz and SNR = 20 dB, and clarify whether these values are representative of practical scenarios.
Response: We thank Reviewer for this insightful comment. The selection of these parameters was indeed based on their practical relevance and common usage in the terahertz imaging research community. Please find our justifications below:
(1) Center Frequency (670 GHz): The frequency of 670 GHz was selected primarily for two reasons. First, it resides within a prominent atmospheric transmission window, which is crucial for practical standoff imaging applications. Second, 670 GHz represents a technologically mature and accessible band within the sub-terahertz regime. Operating at this relatively high frequency is essential for achieving high resolution in TCAI, while still being feasible with current component technology.
- Tian, Y. L.; Li, R. X.; Huang, K.; Deng, X. J.; Jiang, J.; He, Y. A 670 GHz Solid-State Multiplied Source with > 5mW Output Power Based on the Four-Port Prototype and 3D-Stacked Power-Combined Concept. J. Infrared, Millimeter, Terahertz Waves 2025, 46(6), 38.
- Zhang, Z. Q.; Liu, W. X.; Wang, J. L. Study on Electron Optics System for 670 GHz Travelling Wave Tube. 2023 Photonics & Electromagnetics Research Symposium (PIERS), Prague, Czech Republic, 2023, pp. 141-144.
(2) Signal-to-Noise Ratio (SNR = 20 dB): An SNR of 20 dB is a typical and representative value assumed in many theoretical and simulation-based studies of terahertz imaging systems, including:
- Liu, X. Y.; Luo, C. G.; Gan, F. J.; Wang, H. Q.; Peng, L.; Wang, Y. Antenna Phase Error Compensation for Terahertz Coded-Aperture Imaging. Electronics 2020, 9, 628.
- Chen, S. Research on Three-Dimensional Terahertz Coded-Aperture Imaging Technology. Ph.D. Thesis, National University of Defense Technology, Changsha, China, 2018.
- Chen, S.; Luo, C.G.; Deng, B.; Qin, Y.L.; Wang, H.Q. Study on coding strategies for radar coded-aperture imaging in terahertz band. J. Electron. Imaging 2017, 26, 053022.
- Chen, S.; Luo, C.G.; Deng, B.; Qin, Y.L.; Wang, H.Q.; Zhuang Z.W. Research on Resolution of terahertz coded-aperture imaging. Journal of Radars. 2018, 7(1), 127–138.
In summary, the chosen values of 670 GHz and 20 dB SNR are well-justified and representative of practical terahertz imaging scenarios, ensuring the relevance and validity of our simulation results. As suggested, the rationale for the selection of these parameters has been added to the main text.
Comment 2: The manuscript states that TwIST is better than LSM, SBL, and Tikhonov regularization, but the supporting experimental results are limited. Authors should add a comparative figure or table that quantitatively to show the differences of these algorithms.
Response: We sincerely thank Reviewer for this valuable suggestion. We agree that a more comprehensive quantitative comparison further strengthens our conclusion. In direct response to your comment, we have now added Figure 28 to provide a detailed quantitative and visual comparison of the imaging results obtained by LSM, SBL, Tikhonov regularization, and TwIST at L = 8.9333 m, L = 9.8267 m, and L = 11.1 m. The results clearly and consistently demonstrate the superior performance of TwIST in achieving higher reconstruction fidelity and robustness across different conditions, thereby solidifying the claim made in the original manuscript. We believe this new figure offers compelling experimental evidence to directly support our comparative analysis and greatly enhances the discussion in Section 4.2.
Comment 3: In Section 4.2, the discussion of near-field effects for L < 1.6 m is a little bit brief. Authors should discuss the practical significance of the near-field–to–far-field transition region.
Response: We thank Reviewer for this insightful comment. We fully agree that a deeper discussion on the practical implications of the near-field region enhances the value of our study. As suggested, we have significantly expanded the discussion in Section 4.2 to address this point. Specifically, we have
(1) Explained the physical origin of the effects in the near-field region, clarifying that the far-field approximation conditions of L >> |xa – xk| and L >> |ya – yk| break down, invalidating the simplified phase models in Equations (21) and (26). This prevents the phase difference vectors from concentrating or clustering.
(2) Explicitly discussed the practical significance, concluding that operating within this near-field transition region provides a stable operational window that avoids the severe performance oscillations observed at certain integer values of in the far field. This insight offers valuable guidance for application by identifying the imaging range where robust and high-fidelity imaging can be consistently achieved.
We believe these revisions have substantially improved the manuscript by connecting the theoretical observations to their practical application, and we thank Reviewer for prompting this important clarification.
Comment 4: The conclusion section mainly summarizes the findings but lacks forward-looking perspectives. Authors should add a short paragraph to discuss potential future work, such as using optimized coding strategies or adaptive algorithms to mitigate performance degradation. It would be helpful for further enhancement of the application of the study.
Response: We are grateful to Reviewer for this constructive suggestion. As recommended, we have added a new paragraph titled “Future Work” in the Section 5, which outlines potential research directions including deep learning-enhanced reconstruction, advanced sparse sensing methods, and hybrid model-based/data-driven frameworks. This addition provides a forward-looking perspective and enhances the practical implications of our study.
We believe that our manuscript has been significantly improved after addressing all the comments and is now suitable for publication in Sensors. Thank you again for giving us the opportunity to revise our work. We look forward to hearing from you soon.
Thank you and best regards.
Yours sincerely,
Yan Teng

Reviewer 2 Report
Comments and Suggestions for Authors
This manuscript is devoted to analysis of dependence of the terahertz coded aperture imaging (TCAI) performance on the imaging range. Authors revealed the behavior of characteristics of aperture imaging performance in dependence on correlations of components of reference-signal matrix and parameters influenced to these correlations. The non-monotonic degradation of aperture imaging performance with increasing the imaging range at specific integer values of parameter of Lc/f0acaE was revealed in particularly, The topic of manuscript is important for scientific and technical groups in areas of obtaining and processing the THz radar images .
There are some points to make the information more clear or to correct some details:
- It is necessary to rewrite the abstract describing effects and results without formula's designations. The abstract presents the obtained results and observed effects, and describes their essence. The designations of physical quantities and mathematical expressions are usually used in the article itself in the corresponding section. In addition,it is necessary to write "e" and "c" in "ae" and "ac" in the 13th line as subscripts.
- If authors to add the geometry of problem by all designations including L (“imaging range”) in Figure 1 it will be more clear for reading. The meaning of L is clarified in formula (19) only.
- It is necessary to check the subscripts in all text. In some cases, the indexes are the usual size (e.g., ac and bc 177th line, 4th line in Table 1,239th line, ae and be in 6th line in Table 1, 240th line) etc.
- Why does the s’l with subscript “l “ but not “k” appear in (6)? Is this a typo? It is not clear.
- In the matrix equation (8) in the resulting column vector Sr there is an index n for each element Srn. What semantic load does it carry? If it corresponds to the moment of time tn as in (6), then it should change from 1 to N.
- What does the index r in (14) (for δr) in the denominator of formula mean? It is not clear.
- It is necessary to change the font or size of X-axis labels in Figures 5, 8, 18. An orientation may be current (270 degrees angle rotation) but numbers need to be further apart. It's hard to read now.
- Figure 6 are not clear according to colors. Authors demonstrate the correlations near 1 for some elements. The color scale on the left shows red-orange colors near 1. However, the entire image, including the peaks, is gray-black-greenish. And in general the picture is very low-contrast, the peaks are visible only when they go beyond the black square on the white background, corresponding to the plane with a correlation coefficient equal to 0. The same situation with Figure 9. Although greenish shades corresponding to correlations of about 0.4-0.5 can be considered if desired.
- Authors say about designations of the L2-norm and L1-norm after formula (29) (page 14), but the L2-norm designation appeared firstly in formula (16) (page 5).
- What is the value of parameter of Lc/f0acaE in Figure 12e? The (e) L = 17 m is written in Figure caption.
- Authors write “Equation (29) fluctuates, regulated by relaxation parameter η to balance convergence stability and global exploration” (381st -382nd lines. But there is no a parameter η in (29) and further (30,31.32). There is regularization parameter lambda there. Parameter η appears in 381st line. Authors must to clarify this moment.
This manuscript is written sufficiently well and can be published after minor revisions.

Author Response
Manuscript Title: Effect of Imaging Range on Performance of Terahertz Coded-Aperture Imaging
Manuscript ID: sensors-3843017
Authors: Yan Teng, Haodong Yang, Xinhong Cui, Xiaoze Li, and Yanchao Shi
To: Ms. Chelsey Wang
From: ganlong1982@foxmail.com
Dear Editors and Reviewer,
Please find enclosed our revised manuscript entitled “Effect of Imaging Range on Performance of Terahertz Coded-Aperture Imaging” (Manuscript ID: sensors-3843017) for consideration in Sensors.
We would like to express our sincere gratitude to you and Reviewer for the constructive comments and valuable suggestions. These comments have been immensely helpful in improving the quality of our manuscript. We have carefully considered all the points raised and have made significant revisions to the manuscript accordingly.
Below, we provide a point-by-point response to the reviewers’ comments. All changes in the revised manuscript have been highlighted.
Comment 1: It is necessary to rewrite the abstract describing effects and results without formula's designations. The abstract presents the obtained results and observed effects, and describes their essence. The designations of physical quantities and mathematical expressions are usually used in the article itself in the corresponding section. In addition, it is necessary to write "e" and "c" in "ae" and "ac" in the 13th line as subscripts.
Response: We sincerely thank Reviewer for this critical suggestion. We have thoroughly revised the abstract to remove all mathematical formula designations. The results and observed effects are now described using qualitative physical explanations to enhance clarity and accessibility for a broad readership, as suggested. Furthermore, we have carefully checked the entire manuscript and corrected the notation for the coding element and imaging grid cell sizes to the proper subscript form ( and ) throughout the text. (Page 1, Abstract; and entire manuscript)
Comment 2: If authors to add the geometry of problem by all designations including L (“imaging range”) in Figure 1 it will be more clear for reading. The meaning of L is clarified in formula (19) only.
Response: We greatly appreciate Reviewer’s suggestion to enhance the clarity of the system geometry. Accordingly, we have modified Figure 1 to explicitly include the imaging range L using a clear double-headed arrow, making its physical meaning immediately apparent to the reader upon first encountering the figure. The detailed dimensions of the coded apertures and the imaging plane grid, including the coding element size and the imaging grid cell, are comprehensively defined in their respective dedicated figures (Figure 2 and Figure 3) as part of the system parameter description. We believe this structure maintains clarity while avoiding overcomplicating the system overview in Figure 1. The physical model and mathematical definitions used in this work are similar as:
- Chen, S.; Luo, C.G.; Deng, B.; Qin, Y.L.; Wang, H.Q. Study on coding strategies for radar coded-aperture imaging in terahertz band. J. Electron. Imaging 2017, 26, 053022.
- Chen, S. Research on Three-Dimensional Terahertz Coded-Aperture Imaging Technology. Ph.D. Thesis, National University of Defense Technology, Changsha, China, 2018.
- Wu, C. Optimal coding design for low-frequency coded aperture imaging. Master’s thesis, National University of Defense Technology, Changsha, China, 2020.
Thank you for this comment, which has undoubtedly improved the readability of our manuscript.
Comment 3: It is necessary to check the subscripts in all text. In some cases, the indexes are the usual size (e.g., ac and bc 177th line, 4th line in Table 1, 239th line, ae and be in 6th line in Table 1, 240th line) etc.
Response: We sincerely thank Reviewer for their meticulous attention to detail. We have conducted a thorough, word-by-word review of the entire manuscript to identify and correct all instances of inconsistent subscript formatting. All physical parameters, including , , , ,and others, have been uniformly formatted with proper subscripts throughout the text, tables, and figures. This includes the specific locations mentioned by Reviewer and all other occurrences in the manuscript. We appreciate this comment, which has significantly improved the consistency and professionalism of our manuscript.
Comment 4: Why does the s’l with subscript “l” but not “k” appear in (6)? Is this a typo? It is not clear.
Response: We sincerely thank Reviewer for catching this typographical error. Reviewer is absolutely correct. The subscript in Equation (6) should indeed be “k” to maintain consistency with the summation index and the definition of the scattering coefficient vector σ’ in Equation (8). This was a oversight during the manuscript preparation. We have corrected “σ’l” to “σ’k” in Equation (6) of the revised manuscript. We appreciate Reviewer’s meticulous reading.
Comment 5: In the matrix equation (8) in the resulting column vector Sr there is an index n for each element Srn. What semantic load does it carry? If it corresponds to the moment of time tn as in (6), then it should change from 1 to N.
Response: We are grateful to Reviewer for this insightful observation that has improved the precision of our manuscript. Reviewer is correct on both counts. The subscript n in Srn indeed denotes the sampling time instant tn, consistent with its definition in Equation (6). We have revised Equation (8) accordingly in the manuscript: The element notation has been changed from Srn to Sr1, Sr2, Sr3, …, SrN which unambiguously indicates the received echo signal at time tn. We thank Reviewer for their meticulous review, which has enhanced the mathematical rigor of our paper.
Comment 6: What does the index r in (14) (for δr) in the denominator of formula mean? It is not clear.
Response: We sincerely thank Reviewer for identifying this typographical error and for prompting this clarification. Reviewer is absolutely correct. In the original manuscript, in the denominator of Equation (14) should be (referring to the nr-th singular value). This has been corrected in the revised manuscript to ensure the formula is consistent and accurate. nR is the actual rank of RSM St (i.e., the total number of its positive singular values, δ1 ≥ δ2 ≥ ⋯ ≥ ≥ ⋯ ≥ > 0). nr is the effective rank (nr < nR), which is the smallest integer such that the ratio of the second-order norms satisfies:
|
, |
(14) |
In essence, while there are nR singular values in total, only the first nr of them are dominant and carry the majority of RSM’s energy or information. The effective rank nr quantifies the number of these principal components required to represent the essential structure of the matrix within a tolerance defined by ξ (a threshold close to 1). We appreciate Reviewer’s meticulous attention to detail, which has helped us improve the accuracy and clarity of our manuscript.
Comment 7: It is necessary to change the font or size of X-axis labels in Figures 5, 8, 18. An orientation may be current (270 degrees angle rotation) but numbers need to be further apart. It's hard to read now.
Response: We thank Reviewer for this valuable feedback regarding the readability of our figures. We agree that the X-axis labels were too densely spaced.
We have regenerated Figures 5, 8, and 18 to address this issue. Specifically, we have increased the spacing between the X-axis tick labels, and slightly reduced the font size to accommodate the increased spacing without overlap. These modifications have significantly improved the clarity and readability of the figures. The updated versions are included in the revised manuscript.
Comment 8: Figure 6 are not clear according to colors. Authors demonstrate the correlations near 1 for some elements. The color scale on the left shows red-orange colors near 1. However, the entire image, including the peaks, is gray-black-greenish. And in general the picture is very low-contrast, the peaks are visible only when they go beyond the black square on the white background, corresponding to the plane with a correlation coefficient equal to 0. The same situation with Figure 9. Although greenish shades corresponding to correlations of about 0.4-0.5 can be considered if desired.
Response: We sincerely thank Reviewer for this critical feedback regarding the color mapping and contrast in our figures. We agree entirely that the previous colormap made it difficult to discern the correlation peaks and overall structure. In direct response to your comment, we have completely regenerated Figures 6 and 9 with the following specific improvements:
(1) Colormap Replacement: We have replaced the previous colormap with a high-contrast, perceptually uniform colormap. This ensures that values near 1 are intuitive and clear.
(2) Enhanced Contrast: The new colormap provides significantly greater contrast across the entire range of correlation values, making the spatial distribution of correlations and the specific high-value peaks immediately visible.
(3) Validation: We have carefully validated that the colors in the plot accurately correspond to the values on the associated colorbar.
The revised figures now accurately and clearly visualize the column correlation structure of the RSM. The peaks near unity correlation are now distinctly visible. We believe the updated figures are a substantial improvement and greatly enhance the clarity of our presentation. Thank you for this suggestion, which has significantly improved the quality of our manuscript.
Comment 9: Authors say about designations of the L2-norm and L1-norm after formula (29) (page 14), but the L2-norm designation appeared firstly in formula (16) (page 5).
Response: We thank Reviewer for this precise observation regarding the placement of mathematical notation explanations. Reviewer is absolutely correct that the L2-norm notation first appears in Equation (16) and should be defined at its first occurrence for clarity. We have revised the manuscript accordingly. The explanation of the L2-norm designation has been moved from its previous location after Equation (29) and is now introduced immediately following its first use in Equation (16). The explanation of the L1-norm designation remains after Equation (29), as this is its first occurrence in the text. This adjustment ensures that all mathematical notations are defined at their point of first use, improving the logical flow and readability of the manuscript. We appreciate Reviewer’s diligence in ensuring the consistency of our notation.
Comment 10: What is the value of parameter of Lc/f0aCaE in Figure 12e? The (e) L = 17 m is written in Figure caption.
Response: We thank Reviewer for this question, which allows us to provide a more detailed explanation of the degradation mechanism at long imaging ranges. We have updated the caption with explicitly state: “ ”.
Because . When L > 10.05 m, the dimensionless criterion , the imaging plane (comprising ME × NE = 45 × 45 scatterers) contains no scatterer pairs with row and column index differences of . Therefore, the strong, discrete correlation peaks observed at integer values of no longer occur. Instead, two alternative degradation mechanisms dominate, as we now more clearly describe in the caption and in Section 4.2: increased correlation between RSM columns mapping to directly adjacent scatterers and phase coverage reduction in wavefront encoding. Both of these effects intensify monotonically with increasing L, leading to the gradual deterioration in image quality shown in Figure 17(k) and (l). We appreciate Reviewer’s comment, which has allowed us to clarify this important transition in the imaging performance.
Comment 11: Authors write “Equation (29) fluctuates, regulated by relaxation parameter η to balance convergence stability and global exploration” (381st -382nd lines. But there is no a parameter η in (29) and further (30, 31, 32). There is regularization parameter lambda there. Parameter η appears in 381st line. Authors must to clarify this moment.
Response: We thank Reviewer for their meticulous reading and for identifying this inconsistency in the description. Reviewer is correct that the relaxation parameter η is an internal algorithm parameter as described in:
- Bioucas-Dias, J.M.; Figueiredo, M.A.T. A new TwIST: Two-step iterative shrinkage/thresholding algorithms for image restoration. IEEE Trans. Image Process. 2007, 16, 2992–3004.
thus not present in the listed equations, and its sudden introduction there was potentially confusing. To clarify the text and eliminate any ambiguity, we have removed the reference to the parameter η from the sentence. The revised text now simply and clearly states: “And nonmonotonic backtracking is then employed to mitigate fluctuations in the objective function per Equation (29), thereby balancing convergence stability and global exploration.” This revision focuses on the purpose of the nonmonotonic backtracking step without delving into implementation-specific parameters that are not central to our discussion. We appreciate this suggestion, which has improved the clarity of our manuscript.
We believe that our manuscript has been significantly improved after addressing all the comments and is now suitable for publication in Sensors. Thank you again for giving us the opportunity to revise our work. We look forward to hearing from you soon.
Thank you and best regards.
Yours sincerely,
Yan Teng

Reviewer 3 Report
Comments and Suggestions for Authors
This paper investigates how the performance of Terahertz Coded Aperture Imaging (TCAI) changes with imaging range and finds that image quality does not degrade steadily but instead shows critical points where it drops sharply. This happens when certain physical parameters align in a way that causes the phase encoding to become less effective, leading to increased correlation in the system’s reference signal matrix. As the imaging range increases further, these correlations grow stronger, particularly between columns representing nearby or dominant scatterers, which harms image reconstruction quality. Among several reconstruction methods tested, TwIST performs best when these correlations affect only non-important (dummy) scatterers. The study suggests strategies like increasing bandwidth or optimizing coding design to maintain high imaging performance.
-
Can the authors provide more practical insights or examples showing how significant the performance drop is at those critical imaging ranges?
-
How sensitive is the system to small changes in imaging range near the critical points—can slight adjustments recover performance?
-
Have these degradation effects been validated in real-world or experimental TCAI setups, or is the analysis entirely simulation-based?
-
Could the authors elaborate more on how the proposed mitigation strategies (e.g., increasing bandwidth) would be implemented in a real system?
Author Response
Manuscript Title: Effect of Imaging Range on Performance of Terahertz Coded-Aperture Imaging
Manuscript ID: sensors-3843017
Authors: Yan Teng, Haodong Yang, Xinhong Cui, Xiaoze Li, and Yanchao Shi
To: Ms. Chelsey Wang
From: ganlong1982@foxmail.com
Dear Editors and Reviewer,
Please find enclosed our revised manuscript entitled “Effect of Imaging Range on Performance of Terahertz Coded-Aperture Imaging” (Manuscript ID: sensors-3843017) for consideration in Sensors.
We would like to express our sincere gratitude to you and Reviewer for the constructive comments and valuable suggestions. These comments have been immensely helpful in improving the quality of our manuscript. We have carefully considered all the points raised and have made significant revisions to the manuscript accordingly.
Below, we provide a point-by-point response to the reviewers’ comments. All changes in the revised manuscript have been highlighted.
Comment 1: Can the authors provide more practical insights or examples showing how significant the performance drop is at those critical imaging ranges?
Response: We thank Reviewer for raising this crucial point. To directly and quantitatively address the question of how significant the performance drop is at critical ranges, we have added a new Table 4 to the manuscript (now placed below Figure 17). Table 4 provides the precise quantitative metrics (PSI and RIE) for each imaging result shown in Figure 17. The data unequivocally demonstrates that at critical imaging ranges where is an integer, the imaging performance deteriorates dramatically.
Comment 2: How sensitive is the system to small changes in imaging range near the critical points—can slight adjustments recover performance?
Response: We appreciate Reviewer’s insightful question regarding the system’s sensitivity near critical points. Our findings indicate that the sensitivity and the ability to recover performance through slight adjustments depend significantly on the imaging range, with a clear transition occurring around L = 8.71 m.
(1) For L < 8.71 m: The system is highly sensitive, and performance is easily recoverable. In this range, the imaging quality of the TwIST oscillates violently with the imaging range L (Figure 15), as it is exquisitely sensitive to whether the value of is an integer. However, this also means that performance can be fully recovered with minimal adjustments. A slight change in L, sufficient to make a non-integer, immediately restores high-fidelity imaging. For example, at L = 4.9133 m with , the column correlation is close to unity (Figure 24(a)), leading to PSI = 0.5799 < 1 (Figure 15), which constitutes a failed reconstruction. With a minimal increase to L = 5.0 m with , the imaging quality recovers dramatically with PSI = 8.7623, yielding a clear, high-quality image as shown in Figure 17(f).
(2) For L > 8.71 m: The system becomes robust and performance is stable. Because the columns of RSM St corresponding to the dominant scatterers (σ′=1) are no longer strongly correlated with others, TwIST consistently achieves successful reconstruction (PSI > 1) across a broad range of L without exhibiting the violent oscillations.
Comment 3: Have these degradation effects been validated in real-world or experimental TCAI setups, or is the analysis entirely simulation-based?
Response: We thank Reviewer for this critical and relevant question regarding the experimental validation of our findings.
(1) Primarily Simulation-Based Analysis: We acknowledge that the analysis presented in this particular study is primarily simulation-based. However, our simulation framework is built upon the well-established physical principles of wave propagation and coherent detection that are widely used for TCAI. The parameters used in our model are carefully chosen to be representative and aligned with those reported in existing experimental TCAI systems, ensuring that our results have a high degree of physical relevance and predictive value.
- Chen, S.; Luo, C.G.; Deng, B.; Qin, Y.L.; Wang, H.Q. Study on coding strategies for radar coded-aperture imaging in terahertz band. J. Electron. Imaging 2017, 26, 053022.
- Chen, S. Research on Three-Dimensional Terahertz Coded-Aperture Imaging Technology. Ph.D. Thesis, National University of Defense Technology, Changsha, China, 2018.
- Wu, C. Optimal coding design for low-frequency coded aperture imaging. Master’s thesis, National University of Defense Technology, Changsha, China, 2020.
(2) Theoretical Gap in Literature: Our work specifically addresses a notable gap in the theoretical understanding of TCAI systems. The effect of the imaging range L on performance has received comparatively little attention, with prior research focusing primarily on coding strategies and reconstruction algorithms. In fact, as noted in citation 25, even the fundamental relationship was previously unexplored beyond a basic conjecture, stating without validation that the spatiotemporal independence of the RSM “should also scale inversely with L”. This underscores that a dedicated theoretical investigation into the role of L was not only lacking but necessary, with experimental studies being even more scarce.
(3) Commitment to Experimental Validation: We completely agree with Reviewer that experimental validation is the essential next step to conclusively confirm these effects in a real-world setup. This is indeed a central objective of our ongoing and future work.
Comment 4: Could the authors elaborate more on how the proposed mitigation strategies (e.g., increasing bandwidth) would be implemented in a real system?
Response: We thank the reviewer for this practical suggestion. The proposed mitigation strategies can be implemented in a real system as follows:
(1) To increase bandwidth, mature radar techniques like linear frequency modulation (LFM), stepped-frequency continuous wave (SFCW), or frequency hopping, can be adopted to introduce frequency diversity and reduce RSM correlation.
(2) For coding optimization, dynamically programmable metasurfaces can be utilized to implement time-varying code patterns and hybrid aperture designs directly countering the periodic phase condition that causes performance degradation.
We believe that our manuscript has been significantly improved after addressing all the comments and is now suitable for publication in Sensors. Thank you again for giving us the opportunity to revise our work. We look forward to hearing from you soon.
Thank you and best regards.
Yours sincerely,
Yan Teng

Reviewer 4 Report
Comments and Suggestions for Authors
The manuscript identifies a fascinating and important phenomenon in TCAI performance. The technical depth and simulation work are strong. However, the current presentation obscures the core physical insights and does not sufficiently articulate its novelty relative to prior art. Addressing these issues will significantly improve the manuscript's impact and clarity, making it a valuable contribution to the field.
- The abstract is overly technical; it should be rewritten to highlight the core physical insight and potential practical implication in simpler terms.
- The introduction does not adequately position this work within the existing literature, making the claim of novelty feel unsubstantiated. The introduction cites many improvements in TCAI (hardware, algorithms like CNN, etc.) but fails to cite any prior work that has specifically investigated the fundamental, range-dependent phase relationships in the RSM. The statement "the impact of imaging range on TCAI performance, however, remains largely unexplored" is a strong claim that requires direct references to the most relevant works to highlight the knowledge gap this paper fills. Surely, some prior analysis of RSM properties must exist.
- The comparison with LSM, SBL, and Tikhonov in Fig. 27 is excellent and adds value. However, the novelty of this comparison is weakened if the choice of these specific algorithms is not justified. Are they the standard benchmarks in the TCAI field? A sentence citing their common use for comparison would help.
- The paper is heavily mathematical, but the physical meaning of the equations is not sufficiently clarified. For example, the relationship between the integer condition f0LC/ac aE and the degradation in spatiotemporal independence is central to the paper, yet the intuition behind why this produces strong correlations is obscured in long derivations.
- Figures could be simplified with clearer captions describing their physical meaning.
- Figures (e.g., Figs. 4–7, 10–12, etc.) show correlation maps, but the manuscript does not adequately connect these results to physical insights accessible to readers outside a narrow specialty. Suggestion: Add simplified schematic diagrams and explanatory text to show why phase clustering at integer multiples causes imaging degradation, and how this differs from conventional radar range-resolution effects.
- Several grammatical errors and awkward phrasings should be corrected (e.g., “developed phase coverage shrinks” → “phase coverage reduction”).
- Ensure consistency in notation (subscripts and variables vary across sections).
Author Response
Manuscript Title: Effect of Imaging Range on Performance of Terahertz Coded-Aperture Imaging
Manuscript ID: sensors-3843017
Authors: Yan Teng, Haodong Yang, Xinhong Cui, Xiaoze Li, and Yanchao Shi
To: Ms. Chelsey Wang
From: ganlong1982@foxmail.com
Dear Editors and Reviewer,
Please find enclosed our revised manuscript entitled “Effect of Imaging Range on Performance of Terahertz Coded-Aperture Imaging” (Manuscript ID: sensors-3843017) for consideration in Sensors.
We would like to express our sincere gratitude to you and Reviewer for the constructive comments and valuable suggestions. These comments have been immensely helpful in improving the quality of our manuscript. We have carefully considered all the points raised and have made significant revisions to the manuscript accordingly.
Below, we provide a point-by-point response to the reviewers’ comments. All changes in the revised manuscript have been highlighted.
Comment 1: The abstract is overly technical; it should be rewritten to highlight the core physical insight and potential practical implication in simpler terms.
Response: We sincerely thank Reviewer for this insightful comment. We agree that the physical insights and practical implications of our findings should be more clearly highlighted in simpler terms. In response, we have thoroughly revised the abstract to:
(1) More clearly articulate the core physical mechanism causing performance degradation. We now explicitly state that the strong column and row correlations in the RSM arise because “the relative phase differences concentrate or cluster into discrete values” at specific ranges. This clustering is concisely measured by a key dimensionless criterion which determines whether the system enters a high-correlation, low-performance regime.
(2) Emphasize the practical implication of our work for system designers and users. The revised abstract concludes by stating that our findings “provide practical guidance for optimizing TCAI system design and operational range selection to avoid performance degradation zones.” This directly addresses the potential application of our research.
By moving technical derivations out of the abstract and into the main text, we have streamlined the narrative to focus on the fundamental cause and its important consequence (the need to avoid specific imaging ranges). We hope the revised version now better communicates the novelty and practical value of our work to a broad audience.
Comment 2: The introduction does not adequately position this work within the existing literature, making the claim of novelty feel unsubstantiated. The introduction cites many improvements in TCAI (hardware, algorithms like CNN, etc.) but fails to cite any prior work that has specifically investigated the fundamental, range-dependent phase relationships in the RSM. The statement "the impact of imaging range on TCAI performance, however, remains largely unexplored" is a strong claim that requires direct references to the most relevant works to highlight the knowledge gap this paper fills. Surely, some prior analysis of RSM properties must exist.
Response: We sincerely thank Reviewer for this critical and constructive comment, which has helped us significantly improve the positioning of our work. We apologize for the oversight in not more precisely defining the knowledge gap relative to prior analyses of RSM properties. We have thoroughly revised the Introduction to address this point.
In the revised manuscript, we now more carefully acknowledge the valuable existing body of work dedicated to analyzing and improving RSM properties under fixed imaging conditions (e.g., through optimized coding strategies [26] or advanced algorithms [18-24]). We then cite the specific work [25] that provides the conventional hypothesis, stating: "Conventional radar theory suggests an inverse relationship between resolution and imaging range [25]. This has led to the natural, yet unexplored, hypothesis that the spatiotemporal independence of the RSM should also scale inversely with L [25]."
Our key contribution, which directly addresses the identified gap, is the discovery and rigorous analysis of a fundamental, imaging range-dependent phase mechanism that governs RSM independence. Contrary to the conventional expectation of a monotonic inverse relationship, we reveal a non-monotonic, periodic degradation of TCAI performance that can be predicted by a dimensionless criterion, which provides the key threshold of the imaging range for the application of TCAI.
Therefore, we have refined our claim of novelty from "largely unexplored" to the more precise assertion that "In contrast, the passive, inherent relationship between the imaging range L and the spatiotemporal independence of the reference-signal matrix (RSM) has received considerably less attention."
We believe the revised Introduction now accurately contextualizes our work within existing literature, properly credits prior research on RSM, and clearly highlights the specific, fundamental knowledge gap that our study fills by uncovering this novel phase-based relationship.
Comment 3: The comparison with LSM, SBL, and Tikhonov in Fig. 27 is excellent and adds value. However, the novelty of this comparison is weakened if the choice of these specific algorithms is not justified. Are they the standard benchmarks in the TCAI field? A sentence citing their common use for comparison would help.
Response: We sincerely thank Reviewer for their positive assessment of the algorithm comparison in Fig. 27 and for this valuable suggestion to strengthen our justification.
Reviewer is absolutely right. We have now added a justification for the selection of these specific algorithms in the revised manuscript (in Section 4.1). The choice of LSM, Tikhonov regularization, and SBL is intentional, as they represent three distinct and foundational classes of reconstruction methods commonly used as benchmarks in computational imaging and, specifically, in the TCAI field [25, 29]:
LSM serves as the baseline non-regularized method, directly inverting the problem. It is included to provide a fundamental performance reference, despite its known instability with noise and ill-posed conditions.
Tikhonov regularization represents the classical L2-norm regularization approach, which stabilizes the solution by introducing a smoothness constraint. It is a standard technique for solving inverse problems.
SBL is a leading sparse reconstruction algorithm that promotes sparsity through a Bayesian framework. It is frequently employed as a state-of-the-art benchmark for comparing performance in problems like TCAI that inherently possess a sparse prior.
By comparing against these three representative algorithms, we aim to comprehensively demonstrate the performance of TwIST across different methodological paradigms. The citations [25, 29] have been added to substantiate their common use as standard benchmarks in related studies.
Comment 4: The paper is heavily mathematical, but the physical meaning of the equations is not sufficiently clarified. For example, the relationship between the integer condition Lc/f0aCaE and the degradation in spatiotemporal independence is central to the paper, yet the intuition behind why this produces strong correlations is obscured in long derivations.
Response: We sincerely thank Reviewer for this insightful and crucial comment. We fully agree that elucidating the core physical intuition is paramount. We have thoroughly revised the manuscript to address this point by implementing the following two major changes:
(1) Addition of a New Summary Section on Physical Insight: Most importantly, we have added a new Section 3.3 entitled "Summary of Unified Physical Insight". This standalone section is dedicated to explaining the physical meaning and intuition that Reviewer rightly found obscured. It synthesizes the findings from the analyses of both column and row correlations under a unified framework. We explicitly describe the integer criterion as a cause leading to phase ambiguity (where path differences equal integer multiples of the wavelength, making phases differing indistinguishable). We then clearly explain how this fundamental wave phenomenon:
①Causes spatial aliasing, making distinct scatterers appear identical to the system and leading to strong column correlations.
②Causes a loss of coding diversity, making random temporal patterns predictable and leading to strong row correlations.
(2) Simplification of Mathematical Derivations: As suggested, we have significantly streamlined the mathematical derivations in Sections 3.1.2 and 3.2.2.
We believe these revisions have improved the clarity and impact of the manuscript. The central physical mechanism is now presented upfront in an intuitive manner based on the detailed mathematical proofs, making it accessible to a broader audience while preserving the technical depth. We are grateful for Reviewer’s guidance, which has helped us articulate this novel insight more effectively.
Comment 5: Figures could be simplified with clearer captions describing their physical meaning.
Response: We sincerely thank Reviewer for this valuable suggestion. We agree that clearer figures and captions are essential for conveying the physical insights of our work. In response, we have thoroughly revised all relevant figures and their captions to enhance clarity and emphasize the underlying physical mechanisms. Specifically, we have:
(1) For Figure 4、Figure 5、Figure 6、Figure 7、Figure 8、Figure 18、Figure 20、Figure 21、Figure 23、Figure 24、Figure 26 and Figure 28, simplified the figures by removing redundant or non-essential elements, improving visual clarity and focus on key phenomena.
(2) Enhanced the captions of Figure 3 ~ Figure 30 by adding detailed explanations of the physical meaning behind each figure, including: ① The relationship between phase differences and column/row correlations in the RSM; ② How specific imaging ranges lead to degradation in spatiotemporal independence; ③ The impact of integer vs. non-integer values of on imaging performance.
These revisions ensure that each figure now more effectively illustrates the core physical principles discussed in the manuscript, thereby improving the overall readability and interpretability of our results.
We believe these improvements significantly strengthen the presentation of our findings and better highlight the novelty and practical implications of our study.
Comment 6: Figures (e.g., Figs. 4–7, 10–12, etc.) show correlation maps, but the manuscript does not adequately connect these results to physical insights accessible to readers outside a narrow specialty. Suggestion: Add simplified schematic diagrams and explanatory text to show why phase clustering at integer multiples causes imaging degradation, and how this differs from conventional radar range-resolution effects.
Response: We greatly appreciate this insightful comment, which has helped us significantly improve the clarity and accessibility of our manuscript. In direct response to this suggestion, we have made the following revisions to better explain the physical mechanisms behind the observed phenomena and to highlight the novelty of our findings relative to conventional radar systems:
(1) Added a new schematic diagram (Figure 12) in Section 3.3 to visually illustrate how integer values of Lc/f0aCaE lead to concentration in ϕa(k1,k2) and clustering in ϕk(a1, a2), which consequently degrades the spatiotemporal independence of the RSM and impair imaging quality.
(2) Enhanced explanatory text in Section 3.3 to explicitly describe the physical process by which phase concentration and clustering undermine the effectiveness of random discrete phase encoding, thereby reducing the diversity of the measurement matrix and leading to reconstruction failures.
(3) Added a dedicated discussion in the concluding section (Section 5) contrasting the phase-driven degradation mechanism of TCAI with the conventional radar range-resolution effects.
These additions make the physical insights more accessible to a broader audience and clearly articulate the distinctiveness of our work compared to traditional radar imaging paradigms.
We believe these revisions have substantially strengthened the manuscript and thank Reviewer for this constructive feedback.
Comment 7: Several grammatical errors and awkward phrasings should be corrected (e.g., “developed phase coverage shrinks” → “phase coverage reduction”).
Response: We sincerely appreciate Reviewer’s careful reading and valuable suggestions for improving the manuscript’s language. We have thoroughly reviewed the entire text and corrected grammatical errors and awkward phrasings to enhance clarity and readability. Specifically, we have:
(1) “developed phase coverage shrinks” is corrected into “reduced phase coverage”
(2) “Recent advances in terahertz (THz) radar imaging leverage THz waves’ superior penetration over optical waves and higher resolution than microwaves.” is corrected into “Recent advances in terahertz (THz) radar imaging take advantage of THz waves’ superior penetration over optical waves and higher resolution than microwaves.”
(3) “However, these systems remain constrained by conventional synthetic aperture radar (SAR) and inverse SAR (ISAR) methodologies …” is corrected into “However, these systems are still limited by conventional synthetic aperture radar (SAR) and inverse SAR (ISAR) methodologies …”
(4) “It enables instantaneous high-resolution forward-looking and staring imaging...” is corrected into “It facilitates instantaneous high-resolution forward-looking and staring imaging …”
(5) “Since its proposal, significant efforts have advanced TCAI in both hardware and computation.” is corrected into “Since TACI was first proposed, researchers have made significant advances in both hardware and computation.”
(6) “Following N sampling iterations of the scattered wave field, the received echo signals can be compactly represented as the union of Equation (6) in the matrix form …” is corrected into “Following N sampling iterations of the scattered wave field, the received echo signals can be compactly represented in matrix form as …”
(7) “Target reconstruction essentially lies in solving the non-homogeneous linear system...” is corrected into “Target reconstruction essentially involves solving the non-homogeneous linear system …”
(8) “Conversely, RIE measures the ratio of the reconstruction error to the true scattering coefficients with lower values signifying higher reconstruction fidelity.” is corrected into “Conversely, the RIE measures the ratio of the reconstruction error to the true scattering coefficients. Lower RIE values signify higher reconstruction fidelity.”
(9) “TCAI performance degrades non-monotonically with the imaging range L.” is corrected into “TCAI performance varies non-monotonically with the imaging range L.”
These comprehensive edits have significantly improved the quality of the writing, making the physical insights and technical contributions of our work more accessible to the broader readership of Sensors. We are grateful for this feedback, which has helped us polish the manuscript to a higher standard.
Comment 8: Ensure consistency in notation (subscripts and variables vary across sections).
Response: We thank Reviewer for pointing out this critical issue for the clarity and professionalism of the manuscript. We have conducted a thorough, systematic review of the entire document to ensure complete notational consistency. Specifically, we have:
(1) Standardized the symbols for key physical quantities throughout all sections, figures, and captions. The imaging range is now consistently denoted as L. All vectors and matrices are now typeset in boldface, while scalars and the elements of vectors/matrices remain in non-bold typeface.
(2) Unified the subscripts used for indexing coding elements (consistently a, b), scatterers (k), and sampling instants (n).
(3) Carefully checked and corrected all figure axes, labels, and table entries to ensure they match the notation used in the main text.
We believe these efforts have eliminated all inconsistencies in notation, greatly improving the manuscript’s readability and rigor. We are grateful for this suggestion.
We believe that our manuscript has been significantly improved after addressing all the comments and is now suitable for publication in Sensors. Thank you again for giving us the opportunity to revise our work. We look forward to hearing from you soon.
Thank you and best regards.
Yours sincerely,
Yan Teng

Round 2
Reviewer 3 Report
Comments and Suggestions for Authors
The authors have made significant improvements, and the paper is well-prepared for publication.